# Effect of Purple Neem Foliage as a Feed Supplement on Nutrient Apparent Digestibility, Nitrogen Utilization, Rumen Fermentation, Microbial Population, Plasma Antioxidants, Meat Quality and Fatty Acid Profile of Goats

**DOI:** 10.3390/ani12212985

**Published:** 2022-10-30

**Authors:** Nittaya Taethaisong, Siwaporn Paengkoum, Chatsirin Nakharuthai, Narawich Onjai-uea, Sorasak Thongpea, Boontum Sinpru, Jariya Surakhunthod, Weerada Meethip, Pramote Paengkoum

**Affiliations:** 1School of Animal Technology and Innovation, Institute of Agricultural Technology, Suranaree University of Technology, Muang, Nakhon Ratchasima 30000, Thailand; 2Program in Agriculture, Faculty of Science and Technology, Nakhon Ratchasima Rajabhat University, Muang, Nakhon Ratchasima 30000, Thailand

**Keywords:** purple neem foliage, sunflower oil, rumen fermentation, meat quality, fatty acid profile in meat goat

## Abstract

**Simple Summary:**

Purple Neem foliage has been mostly found in Thailand. This plant contains anthocyanins, which may be useful to ruminants. Therefore, as a result, the purpose of this research was to evaluate the effect Purple Neem foliage as a feed supplement on nutrient apparent digestibility, nitrogen utilization, rumen fermentation, microbial population, plasma antioxidants and meat quality in goats. We observed that giving Purple Neem foliage to goats improved feed intake, nutrient digestion, nitrogen utilization, growth performance, rumen fermentation, microbial population, plasma antioxidant, meat quality and fatty acid profile in meat goat by increasing polyunsaturated fatty acid and antioxidant gene expression in meat goat by improve glyceraldehyde-3-phosphate dehydrogenase (GADPH), glutathione peroxidase (GPX), catalase (CAT), and superoxide dismutase (SOD). As a results, Purple Neem foliage as a ruminant feed supplementation may become a viable alternative feed addition in the future.

**Abstract:**

The purpose of this experiment was to investigate the effect of Purple Neem foliage as a feed supplement on nutrient apparent digestibility, nitrogen utilization, rumen fermentation, microbial population, plasma antioxidants, meat quality and fatty acid profile of goats. Eighteen Boer male goats (approximately 20 ± 2 kg body weight; mean ± standard deviation (SD)) were randomly allocated into three treatments. All goats were fed a 60 d daily feeding with three treatments: (1) control, (2) 3% Purple Neem foliage (PNF) + 3% sunflower oil (SFO) in concentrate, and (3) 6% Purple Neem foliage (PNF) + 3% sunflower oil (SFO) in concentrate. The findings indicate that goat feed containing 6% PNF + 3% SFO in concentrate increased feed consumption, nutrient intake, nutrient apparent digestibility and nitrogen utilization compared to the goat feed at 3% PNF + 3% SFO and the control group. The feeding of goats with 6% PNF + 3% SFO in concentrate resulted in high ammonia nitrogen, BUN, acetic acid, propionic acid, butyric acid, and the total VFA levels were increased at 2 and 4 h after feeding (*p* < 0.01). The individual microbial population with 6% PNF + 3% SFO had higher (*p* < 0.01) total bacteria, higher *Butyrivibrio fibrisolven*, *Fibrobacter succinogenes*, *Ruminococcus albus*, *Ruminococcus flavefacises*, and *Streptococcus bovis*, decreased protozoa and methanogen levels at 2 and 4 h after feeding. The antioxidant in plasma indices varied, with 6% PNF + 3% SFO having higher total antioxidant (TAC), superoxide dismutase (SOD), glutathione peroxidase (GPX), 2, 2-diphenyl-1-picrylhydrazyl (DPPH), and catalase (CAT) antioxidant activity and lower malondialdehyde (MDA) in plasma at 2 and 4 h after feeding. Additionally, goat fed 6% PNF + 3% SFO can improve meat quality by lowering drip loss, cooking loss, shear force, and saturated fatty acid as well as increase the fatty acid profile (monounsaturated and polyunsaturated fatty acids) in goat meat. Our findings suggest that Purple Neem foliage might be an excellent alternative additive for goat feed.

## 1. Introduction

Purple Neem (PN) (*Azadirachta indica* A.) is one of two Meliaceae neem cultivars prevalent across Southeast Asia, including Thailand. According to reports, the phenotypical characteristics of the Siamese neem tree are less branching, longer and thicker leaflets, larger and denser inflorescence, and larger fruit [1]. As a result, Purple Neem foliage (PNF) has a complex of various constituents including rich sources of plant secondary compounds, flavonoids, and polyphenolic components. The Siamese neem tree is high in antioxidants and has antifungal and antibacterial properties [1]. Neem leaves are reported to contain a high level of polyphenolic compounds. In recent years, in Thailand, PN has been recognized for its rapid growth, high amounts of foliage, high seed output, high protein content, and abundance of anthocyanins, as well as its ability to tolerate low fertilizer levels and be readily harvested or plowed into soil. Moreover, PNF may have antimicrobial activity, particularly in protozoal and methanogen populations [2]. PNF may influence digestion and rumen fermentation by modifying bacterial growth due to its capacity to interact with fiber and protein content. Therefore, Purple Neem foliage (PNF) is appropriate for usage in ruminant feed ingredient.

The scientific community is becoming more interested in the effects of PNF on animal nutrition. Purple Neem foliage has been described as having anthocyanin content. The functional properties of anthocyanins as beneficial coloring compounds have been studied in animals; it modulates the alleviation of heat stress, accelerates antioxidant activity, and stimulates the rumen microbiome [3,4,5]. Natural plant ecochemicals such as polyphenols, anthocyanins, tannins, and essential oils have been demonstrated to influence rumen microbial populations, while suppressing methanogen, increasing animal performance [2,6], and stimulating the rumen microbiome [3]. Several previous studies [7,8] have indicated that rumen fermentation enhancement and animal productivity as well as reduced methane production and nutritional stress, such as bloating or acidity, promote health. Therefore, this has potential to improve rumen-derived products, such as milk and meat. Purple Neem foliage possesses antioxidant action through inactivating lipid free radicals or inhibiting hydroperoxide breakdown into free radicals. It has been shown that the de-crease in DPPH radical absorbance mediated by phenolic compounds is due to the interaction between antioxidant molecules and radicals [9]. The chemistry and pharmacology of Purple Neem foliage have potential in this context, but they should be extensively examined before such compounds can be recommended for widespread use. Feeding Purple Neem foliage to growing goats reduces not only environmental problems caused by residue accumulation but also the carbon footprint of animal production; because of the presence of bioactive ingredients, it also increases meat shelf-life stability and quality, for example anthocyanins, and the effect of phenolic compounds as natural antioxidants in reducing oxidative deterioration of meat during meat production [10]. The natural antioxidants derived from Purple Neem foliage are regarded as key dietary components engaged in human health promotion and displaying antioxidant activity [11]. Feedback from farmers indicates that feeding PNF alone cannot completely meet ruminant production requirements due to the reduced palatability of anthocyanins and lower dry matter intake. However, this study uses sunflower oil to mix in with the concentrate to improve the palatability of anthocyanins and improve animal performance. Several previous studies [12] have indicated that neem (*Azadirachta indica*) can be used as a protein supplement for goat production. However, as Purple Neem foliage has a bitter taste, the use of sunflower oil (SFO) as an additive increases the feed intake and palatability of the ruminant. Here, SFO has beneficial properties as a source of fatty acids and as a good source of fatty acids and to convert monounsaturated fatty acids (MUFAs) to polyunsaturated fatty acids (PUFAs) or inserting new unsaturated links into previously existing unsaturated bonds to improve products, such as meat and milk [13,14].

Purple Neem foliage has antioxidant properties due to anthocyanins, improves rumen fermentation, and improves microbial populations, and PNF can be added to goat diets with no negative effects on animal health. Based on this, we hypothesized the effect of Purple Neem foliage as a feed supplement on nutrient apparent digestibility, nitrogen utilization, rumen fermentation, microbial population, plasma antioxidants, meat quality and fatty acid profile of goats.

## 2. Materials and Methods

### 2.1. Dried Purple Neem Foliage

The fresh Purple Neem foliage of (*Azadirachta indica* A.) were collected from Suranaree University of Technology (SUT) goat and sheep research farm in Nakhon Ratchasima, Thailand, during November 2020. Authentication of plant material was carried out at the Animal Technology and Innovation, Institute of Agricultural Technology, Suranaree University of Technology laboratory. The preservation process of Purple Neem foliage recommended by [15], was conducted until being offered to the animals.

### 2.2. Location, Experimental Design, Animal Diets and Managements

The testing was endorsed by the Suranaree University of Technology’s Animal Care and Use Committee, (SUT 4/2558), Nakhon Ratchasima province, Thailand. The Suranaree University of Technology (SUT) goat and sheep research farm in Nakhon Ratchasima, Thailand, hosted all of the goats utilized as rumen donors. The research was carried out in accordance with Thailand’s National Research Council’s recommendations on animal experiments and the Animal Welfare Guidelines for Research (B-31/2562).

The treatments were of a completely randomized design (CRD). This study had 3 treatments: T1, control; T2, 3% Purple Neem foliage + 3% sunflower oil in concentrate; T3, 6% Purple Neem foliage (PNF) + 3% sunflower oil (SFO) in concentrate. Eighteen Boer male goats received the 3 dietary treatments, and each treatment had 6 replications. During the 60-day experimental period, all goats were maintained in individual feeding pens. Mineral blocks and clean water were provided ad libitum to all animals. During the experimental period, the goats were fed 1.5% of BW DM/day containing roughage source of Pangola (*Digitaria eriantha*) hay and 16% crude protein at the ratio of 60:40. All goats were fed and supplemented with their specific treatment diets made in compliance with NRC standards. The goats were fed in the morning and afternoon at around 07.00 and 16.00. All animals were fed the same diet throughout the adjustment phase (14 days before the start of each experimental session). Samples were collected during the final 7 days of the experiment. The nutrient composition of the experimental diets is shown in Table 1 and fatty acid composition of the experimental diets is shown in Table 2.

### 2.3. Chemical Composition

The experimental periods lasted 60 d, starting after 14 days of adjustment for the animals and their digestive tracts. Each day, samples of given and refused feeds were collected; during the final 7 d of the study, urine and feces samples were collected from each individual goat. Basal feed 500 g was dried for 72 h in a vacuum oven at 65 °C before being ground in a Wiley mill (Retsch SM 100 mill; Retsch Gmbh, Haan, Germany) with a 1 mm sieve. Chemical and nutritional analyses were performed on the dried samples. The neutral DM, ash, ether extract (EE), and crude protein (CP) content was deemed appropriate [16], along with the NDF and ADF [17]. In addition, the nitrogen content of the two experimental diets was determined, and dietary refusals were assessed using a KjeltecTM 8400 fully automated Kjeldahl analyzer (FOSS, Hillerd, Denmark); the conversion factor used to derive crude protein (CP) values was 6.25. Furthermore, the second subsample was extracted for 24 h at 50 °C with 0.01 N hydrochloric acid (HCl) dissolved in 80% methanol, and the supernatant was collected and transferred to a 50 mL volumetric flask for HPLC determination of anthocyanins [18,19,20]. The chromatographic separation was carried out for 65 min at 28 °C using a reversed-phase column Zorbax SB-C18 column (3.5 m particle size, i.d. 4.6 mm 250 mm, Agilent Technologies, Santa Clara, CA, USA). The injection volume was set to 20 µL. The HPLC-grade acetonitrile and 10% acetic acid were used to make the mobile phase (1:9). For chromatographic separation, a binary gradient of (A) 10% acetic acid, 5% acetonitrile, and 1% phosphoric acid and (B) acetonitrile was used at a flow rate of 0.8 mL/min. The 520 nm photodiode array UV detector was used to measure the absorption of the compounds tested and evaluated.

### 2.4. Feed and Fecal Sampling

The amount of feed delivered and the number of refused samples were recorded on a daily basis throughout the trial. At the conclusion of the time, feed, refusals, and fecal samples were collected from each individual goat utilizing the full collection procedure. Dry matter (DM), ash, ether extract (EE), and crude protein (CP) were evaluated using [5,16]. Acid detergent fiber (ADF) and neutral detergent fiber (NDF) were determined according to [21,22].

### 2.5. Urine Sampling Procedures

On the same days as feces, total urine samples were collected in a plastic container treated with 10% sulfuric acid to keep the final pH below 3 and avoid nitrogen (N) loss. At the end of each period, urine samples were collected at about 100 mL of total urine volume, stored, and pooled for total N analysis utilizing [23].

### 2.6. Apparent Digestibility

The acid insoluble ash (AIA) approach was used to calculate apparent nutrient digestibility (%) as follows: apparent nutritional digestibility (%) = 100 − ((100 × % AIA in diet × % AIA in fecal)/ (% AIA in fecal × % AIA in diet) [24]. All fecal samples were crushed and passed through a 1 mm filter after being oven dried at 65 °C for 72 h before being stored at 4 °C until analysis.

### 2.7. Plasma Antioxidant Enzyme Activity Analysis

Blood samples were drawn through jugular venipuncture into a single 10 mL heparin containing vacuum tube at 0, 2, and 4 h following the morning meal on the last feeding week at 08.00. After centrifugation (Allegra R X-30R Centrifuge, Beckman Coulter, Life Sciences Division Headquarters 5350 Lakeview Parkway S Drive Indianapolis, In 46268, United States) at 3500× *g* for 20 min at 4 °C, the blood sample was transferred to a 1.5 mL tube and stored at 20 °C until the antioxidant activity enzymes in the plasma were tested. The levels of total antioxidant (TAC), superoxide dismutase (SOD), glutathione peroxidase (GPX), malondialdehyde (MDA), 2, 2-diphenyl-1-picrylhydrazyl (DPPH) scavenging capacity and catalase (CAT) in the plasma were determined using commercial kits (Sigma-Aldrich, USA). The product codes were MAK187, MAK379, MAK437, MAK085, MAK088, and MAK381. All measurement procedures were carried out in accordance with the manufacturer’s specifications. As indicated in a technical report, we used an automated enzymatic colorimetric technique on a microplate (96 wells, UV plate), quadruplicate, integrated into a microreader (Varioskan-LUX multimode microplate reader, Thermo Fisher Scientific, Waltham, MA, USA) [25].

### 2.8. Rumen Fermentation Parameters

At the end of this study, ruminal fluid was collected using a stomach tube connected to a vacuum pump at 0, 2, and 4 h after feeding. The pH of the rumen fluid samples was instantly measured with a portable pH meter, and the samples were under anaerobically filtered through four layers of cheesecloth. The ruminal fluid samples were then divided into three portions. The first portion was used for community DNA extraction for estimation of microbial populations. The second portion of the ruminal fluid samples was used for measuring volatile fatty acids, and the third portion of the ruminal fluid samples was used for NH_3_-N measurement using the Kjeldahl method according to [16]. The strained rumen fluid was then placed in a sterile thermos flask and brought immediately to the laboratory. When the filtered rumen fluid arrived at the laboratory, it was separated into two aliquots. The first aliquot (5 mL) of the filtrates was treated with 0.5 mL of 50% (*v*/*v*) HCl, 0.5 mL of a metaphosphoric acid solution (187.5 g/L), and 0.5 mL of a formic acid (250 mL) solution. Gas chromatography (Agilent 6890 GC, Agilent Technologies, Santa Clara, CA, USA) with a silica capillary column (30, 250, 0.25 m) was used to determine the amounts of VFAs in the filtrates. The initial temperature was 40 °C for 2 min, increasing to 100 °C at a rate of 3.5 °C/min and then to 249.8 °C at a rate of 10 °C/min. The total run time was 30 min. The temperature of the boil room was 250 °C; the carrier gas was He (99.99%). The pressure before columniation was 31.391 psi, the carrier gas flow rate was 3.0 mL/min, and the solvent delay time was 3 min.

### 2.9. DNA Extraction and Real-Time PCR Quantification

The rumen fluid mixture was used for quantitative analysis of the microbial population. The ruminal fluid sample was extracted for community DNA, determined using a commercially available kit method (QIAGEN, Inc., Hilden, Germany). The rumen microbe population studied in this experiment included total bacteria, methanogen, and protozoa. *Butyrivibrio fibrisolvens*, *Fibrobacter succinogenes*, *Ruminococcus flavefaciens*, *Streptococcus bovis*, and *Ruminococcus albus* were studied using the real-time PCR technique. The primer set was created to detect the microbial population using real-time PCR. Table 3 shows the sequences of all the primer sets. Six sample-derived standards were created using the treatment pool set of community DNA. For each real-time PCR test, ordinary PCR was utilized to establish sample-derived DNA standards. After that, the PCR product was purified using the QIA fast PCR purification kit (QIAGEN, Inc., Hilden, Germany). The length of the PCR product and the mass concentration were used to calculate the copy number concentration for each sample-derived standard [26]. In total, five real-time PCR standards were created. In the assays of total bacteria, methanogen, and protozoa, the following cycle settings were used: initial denaturation with one cycle of 94 °C for 2 min, followed by 55 cycles of 94 °C for 30 s, 63 °C for 40 s, and 72 °C for 35 s for primer annealing and product elongation. The conditions for *Butyrivibrio fibrisolvens*, *Fibrobacter succinogenes*, *Ruminococcus flavefaciens*, *Streptococcus bovis*, and *Ruminococcus albus* were as follows: 2 s at 94 °C for denaturing, 30 s at 59 °C for annealing, and 1 min at 68 °C (60 cycles) for extension. Each sample was obtained in triplicate, and the PCR findings were pooled and analyzed using 2% agarose gel electrophoresis. To purify the PCR data, an Axy Prep DNA Gel Extraction Kit was employed (Axy gen Bio-sciences, CA, USA). The PCR products were quantified using a Quantus TM Fluorometer after each sample and were combined in an identical ratio and equally swirled according to the sequencing quantity. The Ct values were translated into normalized relative numbers, which corrected for PCR efficacy, using the Light Cycler 480 software version 1.2.9.11 (Roche Applied Science, Basel, Switzerland).

### 2.10. Meat Quality Characteristics Analysis

#### 2.10.1. pH and Color Measurement

The pH value of the meat goat was measured using a portable pH meter (tes-to pH 2, Germany) at 1 and 24 h postmortem; the pH meter was recalibrated after every six samples. For the statistical analysis, the average value obtained by inserting electrodes into three separate points of each sample was considered.

In this investigation, the color of the meat goat was measured on the cut surface with a Minolta Chromameter (Minolta CR-400, Tokyo, Japan) connected with a D65 illuminant and a two-degree standard observer. In the CIELAB color space model, the results were stated in terms of L* (lightness), a* (redness), and b* (yellowness) [27].

#### 2.10.2. Drip Loss, Cooking Loss and Shear Force Determination

The drip loss of the meat in this study was calculated according to the protocol followed by [28]. Meat portions (2.5 cm thick) were weighed, placed in a container with a plastic bag and top, hung for 24 h at 4 °C, and then reweighed. The drip loss was calculated as the percentage ratio of the initial weight, W1 = meat weight before hanging chilled at 4 °C, W2 = meat weight after hanging chilled at 4 °C using the following formula:Drip loss = (W1 − W2)/W1 × 100%

Cooking loss was determined as each sample of the meat was weighed, placed in a vacuum-sealed plastic bag and cooked for 1 h in a water bath maintained at 80 °C [29]. After cooking, the samples were withdrawn from the water bath and chilled for 30 min under running water. The samples were then removed from the bag, wiped dry, and weighed. The cooking losses were computed as a percentage of the original weight, M1 = meat weight before cooking, M2 = meat weight after cooking using the formula: Cooking loss = (M1 − M2)/W1 × 100%

Shear force was measured using the Warner–Bratzler shear force method. The sample of meat goats, cooked as described above, was cut into rectangular cross-section strips (3 × 1.5 × 1.5 cm). All components were sheared using an Instron 5543 with a Warner–Bratzler shear and a crosshead speed of 200 mm/min [30]. Each sample’s sheer force was tested three times and reported in Newtons (N).

#### 2.10.3. Proximate Composition Analysis

Approximately 50 g of the minced meat sample was measured according to the method of the AOAC [17]. The moisture, crude fat, crude protein, and crude ash contents of goat meat was determined using [19]. The protein was measured using a Kjeldahl K9840 analyzer (FOSS, Hillerd, Denmark); the conversion factor used to produce crude protein (CP) values was 6.25. To determine the proximate composition of GM, the separable fat and connective tissue were manually removed and thoroughly ground and homogenized with an Ultra Turrax homogenizer (IKA Werke, GMBH & Co., KG, Staufen, Germany). The moisture, crude fat, crude protein, and crude ash levels of goat meat were estimated according to [31].

#### 2.10.4. Fatty Acid Analysis in Meat

Fatty acids were extracted using the method as described by [32]. Briefly, 5 g of homogenized goat meat was blended with a mixture of chloroform and methanol (2:1, *v*/*v*). Methylation was performed in duplicate according to IUPAC method [33]. The contents were then vortexed in water bath 55 °C for 1.5 h before being combined with 0.58 mL of 24 N sulfuric acid (H_2_SO_4_) in water and chilled at room temperature. After 1.5 h, the mixture was cooled and thoroughly mixed with 3 mL of hexane solvent before being centrifuged. For analysis, 1 mL of supernatant (fatty acid methyl ester, FAME) was put into an injection vial. The FAME was measured using a gas chromatograph (GC-6890 N, Agilent Technologies, Wilmington, NC, USA), a detector (flame ionization), and a separate SP-2560 type column (capillary) (60 m, 2.5 cm, 0.25 m, Supelco Inc., Wilmington, NC, USA). Bellefonte, PA, USA. The sample split ratio was 30:1. The gas chromatographic conditions were set in accordance with [34]. The fatty acids were identified using the retention time ratio to the FAME standard mixture (FAME 37 component, Sigma-Aldrich Co., St. Louis, MI, USA). To identify the fatty acids, the retention time ratio to the FAME standard mixture (FAME 37 component, Sig-ma-Aldrich Co., St. Louis, MI, USA) was utilized.

### 2.11. Statistical Analyses

The collected data were analyzed using one-way analysis of variance (ANOVA) by SAS version 9.1.3 (SAS Inst. Inc., Cary, NC, USA) model: Y*_ij_* = µ + τ*_i_* + ε*_ij_*, where Y*_ij_* is the observation *j* (*j* = 1 to 6) in the treatment, *i* (*i* = control, 3% PNF + 3% SFO, 6% PNF + 3% SFO), µ is the overall mean, τ*_i_* is the effect of the treatment (denoted an unknown parameter), and ε*_ij_* is the random error with a mean of 0 and variance σ^2^ in treatment. Differences between treatment means were determined by Duncan’s new multiple range test [35]. The model included the treatment and sampling time (at 0, 2, 4 h), which were used to specify variations using ANOVA. The relative mRNA abundance was calculated using the data’s average abundance of the gene as a calibrator, and the data were analyzed using ANOVA. When the *p* Value was (*p* < 0.05), means were split using Tukey’s multiple comparisons adjustment.

## 3. Results

### 3.1. Feed Intake and Nutrient Intake

Table 4 shows the effects of varying quantities of Purple Neem foliage mix with sunflower oil on feed intake and nutrient intake. There were highest on feed intake (gDM/d, % BW, and g/kgBW^0.75^) and nutrient intake in terms of OM, CP, and EE (*p* < 0.01) with the 6% PNF + 3% SFO goat feed diet.

### 3.2. Nutrient Apparent Digestibility

There was a significant difference of nutrient apparent digestibility (*p* < 0.01), as indicated by Table 5. Goats fed 6% PNF + 3% SFO had the highest levels of DM, OM, CP and EE. However, in terms of feeding, 6% PNF + 3% SFO resulted had lower levels of NDF and ADF. 

### 3.3. Nitrogen Balance

As shown in Table 6, there was a significant difference of nitrogen utilization parameters (*p* < 0.01). Among all treatments, goats fed 6% PNF + 3% SFO had the highest levels of N intake (g/d), N excretion from urine (g/d), N digestion (g/d), N digestion (%), N retention (g/d), and N retention (%).

### 3.4. Rumen Fermentation Parameter

The data from the rumen fermentation parameters are shown in Table 7 and Table 8, including pH, ammonia nitrogen (NH_3_-N), blood urea nitrogen (BUN), acetic acid (C_2_), propionic acid (C_3_), butyric acid (C_4_), acetic acid: propionic acid (C_2_/C_3_), and total VFA (TVFA) level. The pH, NH_3_-N, BUN, C_2_, C_3_, C_4_, and TVFA were not different between the groups at 0 h (*p* > 0.05). There were significant differences (*p* < 0.01) in NH_3_-N, BUN, C_2_, C_3_, C_4_, C_2_/C_3_, and TVFA among treatments after 2 and 4 h of feeding. Goats receiving 6% PNF + 3% SFO had lower acetic acid: propionic acid values (*p* < 0.01) than the other treatments. In terms of feeding, the 6% PNF + 3% SFO resulted in the highest values of ammonia nitrogen, blood urea nitrogen, acetic acid, propionic acid, butyric acid and total VFA level.

### 3.5. Microbial Population

The effects of Purple Neem foliage mixed with sunflower oil on the microbial population are shown in Table 9. There were significant variations in the microbial community after 2 and 4 h of feeding (*p* < 0.01). Goats fed a diet of 6% PNF + 3% SFO showed higher levels of total bacteria, *Butyrivibrio fibrisolven*, *Fibrobacter succinogenes*, *Ruminococcus albus*, *Ruminococcus flavefacises*, and *Streptococcus bovis* (*p* < 0.01). The goats fed 6% PNF + 3% SFO had reduced levels of protozoa and methanogen at 2 and 4 h post feeding (*p* < 0.01).

### 3.6. Antioxidant Activity in Plasma

Table 10 shows the antioxidant activity in plasma. Total antioxidant (TAC), superoxide dismutase (SOD), glutathione peroxidase (GPX), malondialdehyde (MDA), 2, 2-diphenyl-1-picrylhydrazyl (DPPH), and catalase (CAT) levels were significantly different (*p* < 0.01). As a result, at 2 and 4 h after feeding, goats given 6% PNF + 3% SFO showed higher levels of TAC, SOD, GPX, MDA, DPPH, and CAT (*p* < 0.01). Goats fed 6% PNF + 3% SFO had reduced malondialdehyde levels at 2 and 4 h post feeding (*p* < 0.01).

### 3.7. Meat Quality Characteristics

In the investigation, all treatments had no effect (*p* > 0.05) on slaughter weight, carcass weight, carcass (%), skin, white offal, red offal, pH, and meat color of L*, a*, and b* (Table 11).

Drip loss (%), cooking loss (%), and sheer force (N) were significantly lower (*p* < 0.01) in the 6% PNF + 3% SFO group compared to another group (Table 12). The effect of Purple Neem foliage was reported to not affect (*p* > 0.05) the moisture, dry matter, ash and EE of goat meat. The effect of Purple Neem foliage was reported to did not affect (*p* > 0.05) on proximate composition of goat meat (Table 13).

### 3.8. Fatty Acid Profile in Meat Goat

Table 14 shows that saturated fatty acid (SFA), monounsaturated fatty acid (MUFA), and polyunsaturated fatty acid (PUFA) had significant differences (*p* < 0.01). The saturated fatty acid content was also reduced in the goats fed 6% PNF + 3% SFO group. However, goats fed 6% PNF + 3% SFO were significantly higher (*p* < 0.01) in MUFA and PUFA than that of the other treatments.

## 4. Discussion

### 4.1. Feed Intake and Nutrient Intake

In the current research, goats fed 6% PNF + 3% SFO had a high feed intake and nutritional intake, demonstrating that giving an anthocyanin source did not have a harmful effect on goat palatability, and sunflower oil does not have a detrimental effect on feed intake, as oil contains vitamin E, which can promote palatability for goats. Further intake is estimated to be inversely related to forage fiber content, since the slower digesting component increases in proportion to the digestive system volume [21,36,37]. Our findings are similar to those of [38,39], which showed that red maize high in anthocyanins can boost lamb productivity. As a result, polyphenol-enriched feeds, such as anthocyanins, are increasingly utilized as supplements added to solid feed after weaning, potentially leading to decreased oxidative stress and an enhanced immunological response, health status, and meat quality in lambs [40].

### 4.2. Nutrient Apparent Digestibility

For the nutrient digestion in this experiment, the effect of anthocyanins from Purple Neem foliage resulted in high nutrient digestion because Purple Neem foliage has high levels of crude protein. Despite its bitter taste, neem seed cake has been introduced and studied in the composition of livestock diets because it contains of important amino acids, crude protein, fiber, sulfur, and nitrogen [41,42,43]. Because of its high protein content, it may be used in place of groundnuts or soya beans with no negative consequences. According to [44], neem leaves may replace up to 50% of soya bean meal in ruminant diets without affecting feed intake, dry matter, fiber digestibility, or body weight. As a result, the higher intake of Purple Neem foliage may be explained by its enhanced rumen degradability and a rise in the outflow of the Purple Neem foliage cell wall to the abomasum [45,46]. Because goats given Purple Neem foliage may boost their feed intake and nutrient digestion, this allows for more nutrients to be obtained and improves animal performance. Moreover, because sunflower oil is a source of energy, this allows the animal to obtain nutrients and sustain its performance. Some studies [39,47,48,49] have also found that the neem plant possesses antibacterial, antifungal, antiviral, and pesticidal properties, as well as the ability to improve general development and health performance, while having no adverse impact on important organs. According to [44], the effect of neem foliage showed that locally produced shrubs can replace imported feedstuff concentrate as a protein supplement for goat production.

### 4.3. Nitrogen Balance

In this investigation, the results of nitrogen consumption differed significantly among treatments. When compared to other treatments, goats fed 6% PNF + 3% SFO had greater nitrogen utilization of N intake, N urine, N digestion, N digestion (%), N retention, and N retention (%). This might be because of the high CP content of Purple Neem foliage, which has the ability to boost ruminal fermentation and to increase microbial protein synthesis [50]. The increased CP intake may make N accessible for rumen microorganisms to use as an energy source to manufacture their cells. In contrast, the usage of pellets containing royal poinciana seed meal (PEREM) may supply additional nutrients to rumen microbes while also helping the animal host [51,52]. Anthocyanins may aid in the coordinated release of nitrogen and carbohydrates from purple field corn stover, which is responsible for increased microbial efficiency [51]. All the treatments demonstrated good N consumption, indicating that the nutrients in each treatment properly met the goats’ protein maintenance requirements. Our findings are congruent with the findings of [42,53,54], who discovered that feeding polyphenol-rich plants to sheep enhanced nitrogen utilization and CP digestibility. Anthocyanins can decrease rumen fermentation and boost nitrogen absorption by binding to food proteins.

### 4.4. Rumen Fermentation Parameter

In the current study, there was a significant difference in pH at 2 and 4 h after feeding, and goats fed 6% PNF + 3% SFO had reduced pH, results that are similar to [55], which reported that rumen pH decreased as lipid content increased in animals given the maximum dietary lipid content (60 g/kg). According to [56,57], pH values greater than 6.2 have little effect on ruminal fermentation. The rumen pH varied from 6.75 to 7.00 on average, which is optimal for microbial digestion in the rumen [51].

Furthermore, the CP content of the feed frequently affects the ruminal fluid concentration of NH_3_-N [58]. In the current study, goats fed 6% PNF + 3% SFO had higher NH3-N levels at 2 and 4 h compared to other treatments. The NH_3_-N content in this research ranged from 12.43 to 15.49 mg/dL. When the authors of [59] evaluated the impact of dietary fat type and concentration on ruminal fermentation in dairy cattle, they found comparable results to ours. The cows were fed corn silage-based diets, and the results revealed that adding 40 g/kg fat resulted in a higher ammonia content than adding 20 g/kg fat. These findings reveal that the ruminal NH_3_-N concentration was sufficient to support microbial development (5 mg/dL) [60] and was near or above the ideal range of 12–20 mg/dL at 2 and 4 h post feeding [61]. Furthermore, Purple Neem foliage contains CP, and the goats are likely to obtain increased protein as a result. According to [62], nursing dairy cows consuming greater levels of CP from anthocyanin-rich corn silage had a higher level of NH_3_-N concentration in comparison to the silage control group.

The research shows that the Purple Neem foliage mixed with sunflower oil as a feed supplement on BUN is significantly different at 2 and 4 h after feeding. However, we discovered that goats given 6% PNF + 3% SFO had greater BUN when compared to other treatments. This was most likely due to the higher CP content of Purple Neem foliage, which was then followed by a higher ammonia nitrogen concentration. The concentration of BUN in plasma also indicated the animal feed balance. In this investigation, the BUN in plasma ranged from 11.23 to 15.28 mg/dL. These findings reveal that the BUN in plasma was within or above the recommended limit of 10–20 mg/dL for all treatments [63,64].

The volatile fatty acid concentrations in this investigation differed significantly between 2 and 4 h after feeding in terms of acetic acid, propionic acid, butyric acid, acetic acid:propionic acid, and total VFA level. The VFAs are the principal energy sources for ruminant metabolism. Anthocyanins may promote VFA generation by altering gut flora [65]. The acetic acid concentrations were increased in the goats fed 6% PNF + 3% SFO. This is likely due to a higher content and digestibility of NDF and ADF in goats fed 6% PNF + 3% SFO. However, the rumen VFA concentration is affected by feed intake, feeding frequency, and diet composition. In this experiment, the concentration of acetic acid was measured to be 58.64–63.37 molar proportion, %, which was within the usual range. Similar to the current work, in [62], the rumen molar concentration of acetate was discovered to be affected by dietary fat intake. The development of AA is always accompanied by the generation of H_2_ and CO_2_, but the formation of PA necessitates the formation of H_2_ as PA [62].

Propionic acid levels were increased in goats fed 6% PNF + 3% SFO at 2 and 4 h after feeding, because Purple Neem foliage contains a high anthocyanin content, and anthocyanins may influence glucose metabolism in order to supply more energy to ruminants. Similarly, ref. [62] claims that anthocyanins may be able to influence carbohydrate metabolism in order to supply more energy to ruminants by raising the fraction of PA. The synthesis of anthocyanin-rich purple corn silage revealed a higher concentration of anthocyanins, which seemed to be undigested in the rumen [63]. Furthermore, the kind of VFAs generated in the rumen is determined by the substrate fermented, the rumen environment, and the microbial community [64,65]. Enhanced rumen propionic acid levels are associated with increased insulin secretion, fat deposition, and protein synthesis, while blocking lipolysis and protein breakdown [66]. According to the findings of this investigation, the propionic acid concentration in all treatments was within the optimal range of 18–20% [67]. 

In this investigation, the butyric acid levels differed significantly among treatments at 2 and 4 h after feeding. Our findings reveal that goats given 6% PNF + 3% SFO had greater butyric acid levels than other treatments. This study is similar to [68], which showed that adding purple maize pigment or anthocyanins to the goats’ diet increased the levels of butyric acid. It was demonstrated that flavonoid-rich plants might alter this kind of rumen fermentation in goats. This might be because anthocyanins’ hydroxyl groups are the primary groups responsible for the inhibitory effect and, as a result, may disrupt bacterial cell membranes [55]. This might be due to the fact that anthocyanins do not appear to be degraded in the rumen [63].

The acetic acid:propionic acid ratio showed a significant difference between treatments at 2 h after feeding. However, we discovered that goats given 6% PNF + 3% SFO had a higher ratio of acetic acid:propionic acid compared to other treatments, because Purple Neem foliage has high NDF and ADF digestibility, which leads to an increase in the ratio of acetic acid:propionic acid. As a result, cell wall fiber fermentation boosted the AA:PA ratio in the normal corn treatment [68]. Our findings are congruent with those of [69]. They discovered that nursing dairy cows fed anthocyanin-rich maize silage produced more ruminal fluid VFAs than those fed control silage.

Total VFA levels in this research demonstrate a significant difference between treatments at 2 and 4 h after feeding. Our findings showed that goats given 6% PNF + 3% SFO had a greater total VFA level than other treatments. It was demonstrated that flavonoid-rich plants might alter the type of rumen fermentation in goats [70]. As a result, feeding Purple Neem foliage altered the manner of VFA fermentation. VFAs are the primary source of energy for ruminant metabolism. Anthocyanins may promote VFA generation by altering gut flora [71]. Our findings are consistent with those of [67], who observed that feeding dairy goats purple maize pigment or anthocyanin-rich purple corn stover silage increased total VFA levels. Moreover, the substrate fermented, the rumen environment, and the microbial population all influence the kind of VFA produced in the rumen [72]. This might be due to the fact that the hydroxyl groups in anthocyanins are the major groups responsible for inhibitory action and, as a result, may destroy bacterial cell membranes [73].

### 4.5. Microbial Population

The microbial community in the rumen in this experiment showed a significant difference at 2 and 4 h after feeding. In the current study, we discovered that goats fed a diet containing 6% PNF + 3% SFO had a greater microbial population of total bacteria, *Butyrivibrio fibrisolven*, *Fibrobacter succinogenes*, *Ruminococcus albus*, *Ruminococcus flavefacises*, and *Streptococcus bovis*, as well as less protozoa and methanogen, compared with the other treatments. While both additions impacted the general structure of the bacterial population in distinct ways, they had a comparable ruminal environment and fermentation in terms of fermentation acids. Anthocyanins from Purple Neem foliage can improve ruminal cellulolytic activities by favoring the growth of cellulolytic bacteria such as *Fibrobacter succinogenes*, *Ruminococcus albus*, *Ruminococcus flavefacises*, and *Streptococcus bovis*, because Purple Neem foliage appears to have anthocyanin content, and the function of anthocyanins has been reported to modulate microorganisms in the gastrointestinal tract. Furthermore, the fermented substrate, rumen environment, and microbial population are all important factors in determining the kind of rumen microbial population [74]. Our findings are consistent with those of [75], which found that a combination of *Saccharomyces cerevisiae*, Enterococcus lactis, Bacillus subtilis, Enterococcus faecium, and Lactobacillus casei, and their fermentation products (PROB), enhanced the relative abundance of cellulolytic bacteria, including *Ruminococcaceae*, and that supplementation with mangosteen peel powder considerably boosted the cellulolytic bacteria population [76]. Anthocyanins have been demonstrated to inhibit protozoal populations, improve bacterial and fungal populations, produce propionate, a partitioning factor, enhance microbial protein synthesis output and efficiency, and decrease methanogenesis, all of which improve ruminant performance. *Butyrivibrio fibrisolven* in this study was higher in goats fed 6% PNF + 3% SFO at 2 and 4 h after feeding. A previous study has demonstrated that flavonoid-rich plant extracts improve cell wall degradation as well as the productivity and efficiency of microbial protein production [77]. Anthocyanins are also known to influence the composition and number of ruminal bacterial species by inhibiting or enhancing certain species growth. Anthocyanins are also known to impact the composition and quantity of ruminal bacterial species through specific inhibition or selective augmentation of particular species growth. Thus, anthocyanins can influence rumen fermentation characteristics because they have substantial regulatory effects on ruminal bacteria, hence promoting gastrointestinal tract health [13]. Additionally, anthocyanins have strong antioxidant qualities that can protect the organism from peroxidation damage, hence improving the rumen microbiota. As a result, anthocyanins can reduce heat stress and alter the rumen microbiome, resulting in improved rumen health [39]. Anthocyanins are also known to affect the composition and quantity of ruminal bacterial species by inhibiting or selectively enhancing certain species’ growth [78]. In the current study, goats fed a diet of 6% PNF + 3% SFO had lower protozoa and methanogen levels than other treatments because anthocyanins can reduce the number of ruminal ciliate protozoa and increase the flow of microbial protein from the rumen, increasing feed utilization efficiency and decreasing methanogenesis. Our findings were similar to the findings of [77], which found that mangosteen peel supplementation significantly reduced rumen protozoa generation while increasing the populations of the major cellulolytic bacteria and decreasing methanogen levels. As a gut modulator, anthocyanins may promote the development of good anaerobic bacteria while suppressing harmful bacteria [79,80].

### 4.6. Antioxidant Activity in Plasma

Due to the presence of naturally active polyphenol compounds, many fruits and other purple materials have a high antioxidant potential. Anthocyanins are a source of secondary plant metabolites that are powerful natural antioxidants and free radical (FR) scavengers, have a number of critical physiological roles for consumers, and have a wide range of study and application possibilities [81] As a result, anthocyanins can boost the activity of antioxidant enzymes such as superoxide dismutase (SOD), glutathione peroxidase (GSH-Px), and catalase (CAT), which can further suppress FR in milk. In the current study, anthocyanins from Purple Neem had a significant difference on antioxidant activity in plasma with respect to total antioxidant, SOD, GPX, DPPH, CAT, and MDA at 2 and 4 h after feeding in all treatments. However, this study found that goats fed 6% PNF + 3% SFO had higher total antioxidant, SOD, GPX, DPPH, and CAT levels, possibly because the Purple Neem foliage provided a high concentration of exogenous natural antioxidants, resulting in higher antioxidant activity and lower MDA levels in plasma when compared to other treatments. According to [39,82], the additional anthocyanin groups had greater levels of SOD, GSH-Px, and CAT than the no additional anthocyanin groups. According to [83], the lack of substantial increases in antioxidant enzyme activity (SOD and GPx) might be explained by the small ruminant absorption of anthocyanins. Furthermore, anthocyanins have high antioxidant activity. DPPH is a kind of FR that is reduced in an aqueous solution containing an antioxidant [76]. Thus, anthocyanins in plasma can give electrons to DPPH, raising the degree of DPPH’s scavenging action [83]. This study found that goats given 6% PNF + 3% SFO have higher DPPH levels in their plasma. Because of their unique features, anthocyanins have been shown to inhibit lipid oxidation [84]. The following mechanisms have been proposed for anthocyanins’ inhibition of lipid metabolism: (1) anthocyanins can reduce cholesterol production by lowering the expression of 3-hydroxy-3-methylglutaryl coenzyme gene A; (2) anthocyanins can lower the blood apo B- and apo C-III levels; and (3) cholesteryl ester transfer protein can be inhibited by anthocyanins [85]. MDA levels in plasma were found to be lower in goats given 6% PNF + 3% SFO in this investigation. According to [86], mice given black soybean seed coat extract (BSSCE), a rich source of anthocyanins, had a substantial drop in MDA concentrations as well as an increase in antioxidant enzyme (SOD, GPx, and catalase) activities. A study [87] found that feeding anthocyanin-rich corn silage to dairy cows reduced plasma MDA concentrations.

### 4.7. Meat Quality Characteristics

This study found that anthocyanin from Purple Neem foliage had no influence on carcass parameters such as slaughter weight, carcass weight, carcass (%), skin, white offal, and red offal regardless of treatment, because anthocyanin’s function might increase the rate of meat production. Our results are consistent with those reported by [88].

The pH value is connected to pre-slaughtering, which can also affect meat texture and color [89]. The influence of anthocyanin from Purple Neem foliage on pH at 1 and 24 h was not observed to be significant; however, our pH readings were 6.83–7.25. According to the findings of our study, the pH value is within the normal range and has no effect on meat quality, such as color or shelf life. Previous research has indicated that pH values of cashmere goat meat in the current study (range 7.40 to 6.20) are indicative of good carcass quality [70,89] demonstrate that the influence of dietary grape pomace had no effect on pH value among different groups. pH, however, is determined by the species of animal, as well as by the age of slaughter and storage duration.

Meat color has been recognized as the most essential factor because color is connected with freshness; shoppers have identified meat color for evaluating meat quality. Meat color in this study shows that anthocyanin from Purple Neem leaf has a tendency to reduce meat color (a*) and tends to brown the meat, implying that the oxymyoglobin to metmyoglobin conversion stage and the lipid peroxidation interaction were engaged in meat discoloration [90]. Our findings are consistent with those of [70,91], who found that dietary grape pomace had no effect on meat color coordinates (L*, a*, and b*) because anthocyanin, a dietary antioxidant, can be better incorporated into cellular membranes containing oxidation susceptible phospholipids than the superficial contact made by antioxidants added postmortem. The color of meat is determined by a variety of individual elements and their interactions; however, chevon has been observed to be lighter in color and have more redness than lamb, owing to the decreased intramuscular fat content of goat carcasses [92].

In the current experiment, the effect of anthocyanin from Purple Neem foliage on drip loss and cooking loss was considerable across all treatments. However, drip loss and cooking loss were decreased in goats given 6% PNF + 3% SFO compared to other treatments. Cooking loss, according to [93], is more reliant on the final pH and the cooking condition. Similar to [94,95], our investigation indicated that goat muscles had lower cooking loss than lamb muscles. According to the current study, goats fed 6% PNF + 3% SFO had decreased drip loss and cooking loss. Juiciness of meat is strongly connected to intramuscular lipids and moisture content of the meat, and lean meat contains around 75% water [96]. This concludes that anthocyanin is a natural antioxidant that may be transferred to the muscle where, in conjunction with the native defense system, it can counteract the action of pro-oxidants and protect meat water loss, quality, and nutritious value of goat meat.

For shear force, many factors impact muscle value, including diet, stress before and after slaughter, animal age and breed, muscle type, and cooking procedure. It has been observed that changes in collagen fiber organization in connective tissue, as well as a decrease in soluble collagen concentration, are important variables determining shear force value throughout animal development [97]. Tenderness of goat meat is frequently acceptable to consumers [98]. Shear force was shown to be significantly varied across all treatments in this investigation. Previous research has demonstrated that anthocyanin-rich black cane silage (AS) reduces shear force [3]. However, according to the current study, goats fed 6% PNF + 3% SFO have lower shear force compared to other treatments because anthocyanin is a natural phenolic compound with health-improving characteristics, such as their potential as a viable alternative to antibiotics and synthetic growth boosters in the creation of sustainable animal feed, which can serve as an antioxidant and has been demonstrated to be effective in avoiding milk and meat oxidation [99]. Nonetheless, dietary anthocyanin bioactive compounds obtained from Purple Neem foliage may result in a decrease in shear force by enhancing juiciness and meat softness by reducing cooking loss and drip loss. Similar to the findings of [100], treatment with wine grape pomace reduced shear force in the lamb’s longissimus dorsi muscle. The chemical composition of the goat meat in this investigation was within the range of previously published values [76], and as predicted, it was unaffected by dietary treatments. Our findings are consistent with those of [101], who reported no effect on the protein, fat, or moisture content of suckling lamb flesh [102]. When thyme and rosemary were introduced to sheep diets, there were no changes in muscle lamb chemical composition.

### 4.8. Fatty Acid Profile in Meat

Plant polyphenols can modify fatty acid composition by decreasing oxidation processes in unsaturated fatty acids in broilers [103]. Plant polyphenolic flavonoids have been found to protect UFA against oxidants and to activate antioxidant response element (ARE)-mediated gene expression [104]. In the current study, goats fed a diet of 6% PNF + 3% SFO lowered the percentage of saturated fatty acids in goat meat because anthocyanin inhibited the process of biohydrogenation in the rumen. The authors of [19] discovered that flavonoid-rich plants in the diet of dairy goats might reduce milk individual SFA and total SFA concentrations while increasing the UFA profile. In contrast, the content of MUFA and PUFA was greater in goats fed a diet containing 6% PNF + 3% SFO. Consistent with our findings, the authors of [105] proposed that a phenolic-rich plant extract increased MUFA and PUFA concentrations in the *longissimus dorsi* of dairy cows. The anthocyanin addition in ruminant feed benefits in the prevention of USF [106]. Characterizing the individual and combined impacts of particular phytochemicals in Purple Neem foliage with anthocyanin concentration will have biohydrogenation products that might assist in increasing the quantities of human-health-promoting PUFAs in ruminant meat [74,95]. *Piper betle* powder can enhance conjugated linoleic acid accumulation from biohydrogenation products, with more PUFA accumulation in the rumen. The addition of fermented Saccharina japonica and Dendropanax morbifera to meat has been shown to enhance PUFA levels [107]. Furthermore, polyphenols might modulate rumen PUFA biohydrogenation to increase lipid fraction by lowering rumen skatole production, increasing the beneficial fatty acid content, and raising the product’s oxidation stability [40].

## 5. Conclusions

This study found that goats fed a 6% PNF + 3% SFO diet had the potential to improve their feed intake, nutrient intake, nutrient apparent digestibility, nitrogen utilization, rumen fermentation parameters, microbial population, plasma antioxidant activity, meat quality and fatty acid profile of goats. Furthermore, goats fed a 6% PNF + 3% SFO diet increased their rumen fermentation parameters via modulating total bacteria abundance, *Butyrivibrio fibrisolven*, *Fibrobacter succinogenes*, *Ruminococcus albus*, *Ruminococcus flavefacises*, and *Streptococcus bovis* and reducing protozoa and methane production. Antioxidant activity in goat plasma could enhance DPPH scavenging activity, SOD activity, TAC, CAT, and GPX and reduce MDA. Additionally, it can improve meat quality and the fatty acid profile in goat meat. Future research is required to validate the current findings and to examine the impact of Purple Neem foliage on the metabolites and gene expression in meat.

## Figures and Tables

**Table 1 animals-12-02985-t001:** Experimental diet components and chemical composition.

Items	Control	3% PNF + 3% SFO	6% PNF + 3% SFO
soybean meal	18.00	16.00	14.00
Rice bran	32.00	24.00	25.80
cassava chip	28.00	27.00	24.30
corn	20.40	25.40	25.30
salt	0.40	0.40	0.40
limestone	0.20	0.20	0.20
premix	1.00	1.00	1.00
Sunflower oil	0.00	3.00	3.00
Purple Neem	0.00	3.00	6.00
Chemical composition (% DM)			
Dry matter	88.66	88.87	89.02
Ash	6.17	6.95	6.19
Crude protein	16.01	16.20	16.29
Ether extract	4.78	5.44	5.48
Non-fibrous carbohydrate	44.84	45.02	41.22
Neutral detergent fiber	28.20	26.39	30.82
Acid detergent fiber	10.71	13.33	15.92
TDN, %	87.76	84.80	83.84
Metabolizable energy, Mcal/kg DM	3.17	3.07	3.03

Contains per kilogram premix: 10,000,000 IU vitamin A; 70,000 IU vitamin E; 1,600,000 IU vitamin D; 50 g iron; 40 g zinc; 40 g manganese; 0.1 g cobalt; 10 g copper; 0.1 g selenium; 0.5 g iodine. Calculated as: NFC =100 − (% NDF + % CP + % EE + % ash). Estimated by the equation TDN (%) = DCP + DNFC) + DEE × 2.25 + (DNDF). Estimated by the equation ME (Mcal/kg DM) = (TDN × 0.04409 × 0.82).

**Table 2 animals-12-02985-t002:** Fatty acid composition in experimental diet components.

Fatty Acids, g/100 g of Total Fatty Acids	Control	3% PNF + 3% SFO	6% PNF + 3% SFO	SEM	*p* Value
Palmitic C16:0	22.16	22.15	22.19	0.01	0.41
Palmitoleic C16:1	1.35 ^c^	1.45 ^b^	1.65 ^a^	0.03	0.01
Stearic C18:0	2.15 ^c^	4.55 ^b^	6.25 ^a^	0.45	0.01
Oleic C18:1	27.49 ^c^	28.50 ^b^	29.85 ^a^	0.27	0.01
Linoleic C18:2 n-6	27.50 ^c^	28.15 ^b^	29.25 ^a^	0.2	0.01
α-Linolenic C18:3 n-3	2.34 ^c^	2.46 ^b^	2.56 ^a^	0.02	0.01
Total saturated fatty acids	24.31 ^c^	26.70 ^b^	28.44 ^a^	0.45	0.01
Total monounsaturated fatty acids	28.84 ^c^	29.95 ^b^	31.50 ^a^	0.3	0.01
Total polyunsaturated fatty acids	29.84 ^c^	30.61 ^b^	31.81 ^a^	0.22	0.01
n-6: n-3 fatty acid ratio	11.75 ^a^	11.45 ^b^	11.44 ^b^	0.05	0.01

PNF, purple Neem foliage; SFO, sunflower oil; ^a, b, c^ in the same row, there is a statistically significant difference (*p* < 0.05); SEM, standard error of the mean.

**Table 3 animals-12-02985-t003:** PCR primers for real-time PCR.

Items	Product Size (bp)	F/R	Sequence
Total bacteria	130	F	CGGCAACGAGCGCAACCC
		R	CCATTGTAGCACGTGTGTAGCC
Methanogen	140	F	TTCGGTGGATCDCARAGRGC
		R	GBARGTCGWAWCCGTAGAATC
Protozoa	223	F	CTTGCCCCTCYAATCGTWCT
		R	GCTTTCGWTGGTAGTGTATT
*Butyrivibrio fibrisolvens*	64	F	ACACACCGCCCGTCACA
		R	TCCTTACGGTTGGGTCACAGA
*Fibrobacter succinogenes*	446	F	GGTATGGGATGAGCTTGC
		R	GCCTGCCCCTGAACTATC
*Ruminococcus flavefaciens*	295	F	TCTGGAAACGGATGGTA
		R	CCTTTAAGACAGGAGTTTACAA
*Ruminococcus albus*	176	F	CCCTAAAAGCAGTCTTAGTTCG
		R	CCTCCTTGCGGTTAGAACA
*Streptococcus bovis*	82	F	TTCCTAGAGATAGGAAGTTTCTTCGG
		R	ATGATGGCAACTAACAATAGGGGT

**Table 4 animals-12-02985-t004:** Effect of Purple Neem foliage as a feed supplement on feed intake.

Items	Control	3% PNF + 3% SFO	6% PNF + 3% SFO	SEM	*p* Value
Feed intake					
gDM/d	855.60 ^c^	971.56 ^b^	1055.11 ^a^	24.68	0.01
% BW	3.07 ^c^	3.15 ^b^	3.27 ^a^	0.02	0.01
g/kgBW^0.75^	43.96 ^c^	59.04 ^b^	61.64 ^a^	2.09	0.01
Nutrient intake					
OMI, g/d	750.33 ^c^	817.74 ^b^	943.31 ^a^	21.37	0.01
CPI, g/d	81.46 ^c^	96.33 ^b^	98.52 ^a^	2.03	0.01
EEI, g/d	38.79 ^c^	40.57 ^b^	45.62 ^a^	0.77	0.01
NDFI, g/d	696.60 ^a^	573.44 ^b^	556.56 ^c^	16.72	0.01
ADFI, g/d	373.52 ^a^	274.21 ^b^	225.36 ^c^	16.50	0.01

PNF, Purple Neem foliage; SFO, sunflower oil; SEM, standard error of the mean; g DM/d, daily intake of dry matter; % BW, edible quantity per body weight per day; g/kgBW^0.75^, daily consumption per daily metric weight; ^a, b, c^, in the same row, there is a statistically significant difference *p* < 0.05.

**Table 5 animals-12-02985-t005:** Effect of Purple Neem foliage as a feed supplement on apparent digestibility.

Items	Control	3% PNF + 3% SFO	6% PNF + 3% SFO	SEM	*p* Value
Apparent Digestibility, % of intake					
DDM	66.80 ^c^	74.30 ^b^	75.61 ^a^	1.04	0.01
DOM	51.67 ^c^	52.77 ^b^	56.91 ^a^	0.62	0.01
DCP	47.78 ^c^	51.62 ^b^	54.58 ^a^	0.75	0.01
DEE	43.40 ^c^	45.53 ^b^	48.56 ^a^	0.57	0.01
DNDF	64.48 ^a^	63.53 ^b^	62.61 ^c^	0.21	0.01
DADF	62.48 ^a^	60.26 ^b^	59.68 ^c^	0.33	0.01

PNF, Purple Neem foliage; SFO, sunflower oil; SEM, standard error of the mean; DDM, digestibility of dry matter; DOM, digestibility of organic matter; DCP, protein digestibility; DEE, fat digestibility; DNDF, digestibility of neutral detergent fiber; DADF, digestibility of acid detergent fiber; ^a, b, c^ in the same row indicates that there is a statistically significant difference *p* < 0.05.

**Table 6 animals-12-02985-t006:** Effect of Purple Neem foliage as a feed supplement on nitrogen balance.

Items	Control	3% PNF + 3% SFO	6% PNF + 3% SFO	SEM	*p* Value
N intake, g/d	11.08 ^c^	12.74 ^b^	13.54 ^a^	0.28	0.01
N Faces, g/d	4.75 ^a^	3.56 ^b^	3.25 ^c^	0.18	0.01
N Urine, g/d	0.13 ^c^	0.16 ^b^	0.18 ^a^	0.006	0.01
N digestion, g/d	4.06 ^c^	6.53 ^b^	8.46 ^a^	0.49	0.01
N digestion (%)	37.35 ^c^	48.40 ^b^	62.20 ^a^	2.72	0.01
N retention, g/d	4.54 ^c^	6.48 ^b^	8.49 ^a^	0.44	0.01
N retention (%)	33.84 ^c^	47.68 ^b^	61.22 ^a^	3.00	0.01

PNF, Purple Neem foliage; SFO, sunflower oil; SEM, standard error of the mean; ^a b c^ in the same row, there is a statistically significant difference (*p* < 0.05).

**Table 7 animals-12-02985-t007:** Effect of Purple Neem foliage as a feed supplement on rumen fermentation.

Items	Control	3% PNF + 3% SFO	6% PNF + 3% SFO	SEM	*p* Value
pH					
0 h	6.95	6.96	6.93	0.008	0.37
2 h	6.75	6.76	6.77	0.008	0.58
4 h	6.85	6.86	6.87	0.01	0.38
Mean	6.85	6.86	6.86	0.007	0.89
Ammonia nitrogen mg/dL					
0 h	12.43	12.65	12.52	0.07	0.47
2 h	12.84	13.34	14.49	0.2	0.01
4 h	13.24 ^c^	14.34 ^b^	15.49 ^a^	0.25	0.01
Mean	12.84 ^c^	13.44 ^b^	14.17 ^a^	0.15	0.01
BUN mg/dL					
0 h	11.23	11.22	11.24	0.62	0.008
2 h	12.35 ^c^	13.42 ^b^	14.35 ^a^	0.22	0.01
4 h	13.48 ^c^	14.20 ^b^	15.28 ^a^	0.21	0.01
Mean	12.35 ^c^	12.95 ^b^	13.62 ^a^	0.14	0.01

PNF, Purple Neem foliage; SFO, sunflower oil; BUN, blood urea nitrogen; ^a, b, c^ in the same row, there is a statistically significant difference (*p* < 0.05); SEM: standard error of the mean.

**Table 8 animals-12-02985-t008:** Effect of Purple Neem foliage as a feed supplement on volatile fatty acid.

Items	Control	3% PNF + 3% SFO	6% PNF + 3% SFO	SEM	*p* Value
Acetic acid (molar proportion, %)					
0 h	58.64	58.65	58.66	0.03	0.97
2 h	59.74 ^c^	61.07 ^b^	62.29 ^a^	0.28	0.01
4 h	60.12 ^c^	62.08 ^b^	63.37 ^a^	0.36	0.01
Mean	59.50 ^c^	60.60 ^b^	61.44 ^a^	0.21	0.01
Propionic acid (molar proportion, %)					
0 h	23.64	23.67	23.68	0.02	0.76
2 h	24.80 ^c^	25.15 ^b^	26.48 ^a^	0.19	0.01
4 h	25.43 ^c^	26.07 ^b^	27.45 ^a^	0.22	0.01
Mean	24.63 ^c^	24.96 ^b^	25.87 ^a^	0.14	0.01
Butyric acid (molar proportion,%)					
0 h	17.71	17.73	17.74	0.02	0.72
2 h	18.78 ^b^	19.21 ^b^	20.46 ^a^	0.21	0.01
4 h	19.87 ^c^	20.19 ^b^	21.36 ^a^	0.17	0.01
Mean	18.79 ^c^	19.04 ^b^	19.85 ^a^	0.12	0.01
Acetic acid: Propionic					
0 h	2.44	2.45	2.46	0.005	0.35
2 h	2.41 ^a^	2.43 ^b^	2.35 ^c^	0.14	0.01
4 h	2.36 ^a^	2.38 ^b^	2.31 ^c^	0.21	0.01
Mean	2.40 ^a^	2.42 ^b^	2.71 ^c^	0.07	0.01
Total VFA (mmol/L)					
0 h	86.15	86.19	86.21	0.07	0.94
2 h	89.50 ^c^	92.19 ^b^	95.38 ^a^	0.64	0.01
4 h	92.29 ^c^	96.07 ^b^	98.56 ^a^	0.69	0.01
Mean	89.31 ^c^	91.48 ^b^	93.38 ^a^	0.45	0.01

PNF, Purple Neem foliage; SFO, sunflower oil; SEM, standard error of the mean; ^a, b, c^ in the same row, there is a statistically significant difference (*p* < 0.05).

**Table 9 animals-12-02985-t009:** Effect of Purple Neem foliage as a feed supplement on microbial population.

Items	Control	3% PNF + 3% SFO	6% PNF + 3% SFO	SEM	*p* Value
Total bacteria (lg10 copies/mL)					
0 h	4.52	4.63	4.70	0.08	0.66
2 h	4.07 ^c^	6.18 ^b^	7.03 ^a^	0.35	0.01
4 h	4.03 ^b^	6.08 ^a^	6.71 ^a^	0.35	0.0001
Mean	4.21 ^b^	5.63 ^a^	6.15 ^a^	0.25	0.0001
*Butyrivibrio fibrisolven* (lg10 copies/mL)					
0 h	5.40	5.41	5.43	0.07	0.99
2 h	6.15 ^c^	7.26 ^b^	8.29 ^a^	0.26	0.01
4 h	6.11 ^c^	7.22 ^b^	8.09 ^a^	0.24	0.01
Mean	5.89 ^c^	6.63 ^b^	7.27 ^a^	0.16	0.01
*Fibrobacter succinogenes* (lg10 copies/mL)					
0 h	3.59	3.54	3.53	0.08	0.96
2 h	4.60 ^b^	4.50 ^b^	7.32 ^a^	0.40	0.01
4 h	3.54 ^b^	4.05 ^b^	7.23 ^a^	0.44	0.01
Mean	3.73 ^b^	4.03 ^b^	6.03 ^a^	0.28	0.01
*Ruminococcus albus* (lg10 copies/mL)					
0 h	3.59	3.45	3.57	0.07	0.73
2 h	4.46 ^c^	5.25 ^b^	6.26 ^a^	0.22	0.01
4 h	4.26 ^b^	5.08 ^b^	6.15 ^a^	0.24	0.0003
Mean	4.10 ^b^	4.59 ^b^	5.33 ^a^	0.17	0.0018
*Ruminococcus flavefacises* (lg10 copies/mL)					
0 h	3.57	3.58	3.59	0.08	0.99
2 h	4.34 ^c^	6.31 ^b^	8.30 ^a^	0.45	0.01
4 h	4.14 ^c^	6.11 ^b^	8.15 ^a^	0.46	0.01
Mean	4.02 ^c^	5.34 ^b^	6.68 ^a^	0.31	0.01
*Streptococcus bovis* (lg10 copies/mL)					
0 h	4.55	4.56	4.57	0.08	1.00
2 h	5.05 ^c^	6.08 ^b^	7.15 ^a^	0.26	0.0001
4 h	3.92 ^c^	5.09 ^b^	7.09 ^a^	0.37	0.01
Mean	4.51 ^c^	5.25 ^b^	6.27 ^a^	0.22	0.01
Protozoa (lg10 copies/mL)					
0 h	7.58	7.55	7.21	0.09	0.19
2 h	9.08 ^a^	6.29 ^b^	5.24 ^c^	0.45	0.01
4 h	8.03 ^a^	6.14 ^b^	5.03 ^c^	0.35	0.01
Mean	8.23 ^a^	6.66 ^b^	5.83 ^c^	0.29	0.01
Methanogen (lg10 copies/mL)					
0 h	8.33	8.55	8.57	0.08	0.43
2 h	9.38 ^a^	8.14 ^b^	6.94 ^c^	0.29	0.01
4 h	9.34 ^a^	8.06 ^b^	6.93 ^c^	0.28	0.01
Mean	9.01 ^a^	8.25 ^b^	7.48 ^c^	0.19	0.0003

PNF, Purple Neem foliage; SFO, sunflower oil; SEM, standard error of the mean; ^a, b, c^ in the same row, there is a statistically significant difference (*p* < 0.05).

**Table 10 animals-12-02985-t010:** Effect of Purple Neem foliage as a feed supplement on antioxidant activity in plasma of goats.

Items	Control	3% PNF + 3% SFO	6% PNF + 3% SFO	SEM	*p* Value
Total antioxidant (nmol/uL)					
0 h	1.58	1.61	1.64	0.05	0.89
2 h	2.07 ^c^	2.28 ^b^	2.46 ^a^	0.05	0.01
4 h	1.79 ^c^	2.05 ^b^	2.17 ^a^	0.04	0.01
Mean	1.82 ^b^	1.98 ^a^	2.09 ^a^	0.04	0.0005
SOD (inhibition rate %)					
0 h	88.91	88.83	89.04	0.21	0.93
2 h	89.17 ^c^	90.45 ^b^	92.62 ^a^	0.41	0.01
4 h	90.40 ^b^	90.87 ^a^	91.19 ^a^	0.10	0.0003
Mean	89.49 ^b^	90.05 ^b^	90.95 ^a^	0.20	0.0017
GPX (units/mL)					
0 h	67.60	67.41	67.71	0.13	0.66
2 h	72.83 ^c^	74.15 ^b^	76.35 ^a^	0.40	0.01
4 h	67.37 ^b^	69.93 ^a^	70.39 ^a^	0.40	0.01
mean	69.27 ^c^	70.50 ^b^	71.48 ^a^	0.26	0.01
CAT (nmol/min/mL)					
0 h	9.61	9.80	9.79	0.14	0.84
2 h	14.79 ^b^	14.87 ^a^	14.93 ^a^	0.02	0.0003
4 h	13.81 ^b^	13.89 ^a^	13.96 ^a^	0.02	0.0003
Mean	12.73	12.86	12.89	0.05	0.43
DPPH scavenging activity (%)					
0 h	35.43	35.48	35.74	0.55	0.97
2 h	43.37 ^c^	47.79 ^b^	52.92 ^a^	1.16	0.01
4 h	38.99 ^b^	52.14 ^a^	56.11 ^a^	2.10	0.01
Mean	39.26 ^b^	45.14 ^a^	48.26 ^a^	1.10	0.01
MDA (μg/mL)					
0 h	28.04	27.75	27.76	0.27	0.9
2 h	36.07 ^a^	30.50 ^b^	25.13 ^c^	1.30	0.01
4 h	32.14 ^a^	30.13 ^b^	28.42 ^c^	0.43	0.01
Mean	32.08 ^a^	29.46 ^b^	27.10 ^c^	0.60	0.01

PND, Purple Neem foliage; SFO, sunflower oil; SOD, superoxide dismutase; GPX, glutathione peroxidase; MDA, malondialdehyde; DPPH, 2, 2-diphenyl-1-picrylhydrazyl; CAT, catalase; ^a, b, c^ in the same row, there is a statistically significant difference (*p* < 0.05); SEM, standard error of the mean.

**Table 11 animals-12-02985-t011:** Effect of Purple Neem foliage as a feed supplement on carcass characteristics of goats.

Items	Control	3% PNF + 3% SFO	6% PNF + 3% SFO	SEM	*p* Value
Slaughter weight (SW), kg	32.57	32.58	32.61	0.02	0.71
Carcass weight (kg)	17.61	17.62	17.64	0.05	0.97
Carcass (%)	48.46	48.5	48.36	0.07	0.77
Skin g/100 g SW	12.5	12.52	12.56	0.06	0.93
White offal, g/100 g SW	7.50	7.52	7.54	0.05	0.95
Red offal, g/100 g SW	0.44	0.46	0.49	0.01	0.16

PNF, purple Neem foliage; SFO, sunflower oil; SEM, standard error of the mean.

**Table 12 animals-12-02985-t012:** Effect of Purple Neem foliage as a feed supplement on meat quality characteristics of goats.

Items	Control	3% PNF + 3% SFO	6% PNF + 3% SFO	SEM	*p* Value
pH value at					
1 h	7.23	7.24	7.25	0.01	0.44
24 h	6.85	6.84	6.83	0.006	0.79
Mean	7.03	7.04	7.05	0.005	0.9
Meat Color					
L*	31.54	31.63	31.6	0.04	0.72
a*	1.75	1.76	1.77	0.01	0.76
b*	5.27	5.28	5.25	0.01	0.66
Drip loss (%)	5.45 ^a^	4.51 ^b^	3.82 ^c^	0.2	0.0001
Cooking loss (%)	32.79 ^a^	29.32 ^b^	24.44 ^c^	1.04	0.0001
Shear force (N)	11.70 ^a^	8.93 ^b^	7.28 ^c^	0.52	0.01

^a, b, c^ in the same row, there is a statistically significant difference (*p* < 0.05); SEM, standard error of the mean; PNF, Purple Neem foliage; SFO, sunflower oil.

**Table 13 animals-12-02985-t013:** Effect of Purple Neem foliage as a feed supplement on chemical composition of meat goats.

Items	Control	3% PNF + 3% SFO	6% PNF + 3% SFO	SEM	*p* Value
Moisture, (%)	76.6	76.63	76.64	0.01	0.44
Dry matter, (%)	23.4	23.37	23.36	0.01	0.44
CP, (%)	26.65	26.66	26.68	0.01	0.66
Ash, (%)	4.16	4.17	4.18	0.007	0.48
EE, (%)	10.1	10.11	10.13	0.01	0.49

PNF, Purple Neem foliage; SFO, sunflower oil; SEM, standard error of the mean.

**Table 14 animals-12-02985-t014:** Effect of Purple Neem foliage as a feed supplement on fatty acid profile in meat goat.

Fatty Acids, g/100 g of Total Fatty Acids	Control	3% PNF + 3% SFO	6% PNF + 3% SFO	SEM	*p* Value
Saturated FA (SFA)					
C14:0	5.54 ^a^	4.41 ^b^	3.48 ^c^	0.24	0.01
C15:0	0.45 ^a^	0.36 ^b^	0.25 ^c^	0.02	0.01
C16:0	24.71 ^a^	23.72 ^b^	22.68 ^c^	0.23	0.01
C17:0	1.76 ^a^	1.66 ^b^	1.56 ^c^	0.22	0.01
C18:0	19.72 ^a^	18.90 ^b^	17.71 ^c^	0.23	0.01
C20:0	0.31 ^a^	0.30 ^b^	0.20 ^c^	0.01	0.01
C22:0	2.37 ^a^	2.23 ^b^	2.10 ^c^	0.03	0.0001
Other SFA	0.97 ^a^	0.85 ^b^	0.75 ^c^	0.02	0.01
Total SFA	55.83 ^a^	52.43 ^b^	48.73 ^c^	0.6	0.01
Monounsaturated FA (MUFA)					
C16:1 cis-9	1.47 ^c^	1.55 ^b^	1.65 ^a^	0.02	0.01
C17:1 cis-10	1.36 ^c^	1.47 ^b^	1.55 ^a^	0.02	0.01
C18:1 cis-9	15.64 ^c^	16.80 ^b^	18.73 ^a^	0.35	0.01
C20:1 cis-11	1.27 ^c^	1.45 ^b^	1.65 ^a^	0.04	0.01
Other MUFA	0.66 ^c^	0.85 ^b^	1.09 ^a^	0.05	0.01
Total MUFA	20.40 ^c^	22.12 ^b^	24.67 ^a^	0.47	0.01
Polyunsaturated FA (PUFA)					
C18:2 n-6	4.56 ^c^	6.58 ^b^	7.70 ^a^	0.35	0.01
C18:3 n-3	0.95 ^c^	1.25 ^b^	1.45 ^a^	0.06	0.01
C20:3 n-6	0.95 ^c^	1.15 ^b^	1.25 ^a^	0.03	0.01
C20:4 n-6	2.53 ^c^	3.68 ^b^	4.53 ^a^	0.23	0.01
C20:5 n-3 (EPA)	0.84 ^c^	1.05 ^b^	1.15 ^a^	0.03	0.01
C22:6 n-3 (DHA)	0.35 ^c^	0.55 ^b^	0.65 ^a^	0.03	0.01
Other PUFA	0.41	0.35	0.45	0.08	0.1
Total PUFA	10.59 ^c^	14.61 ^b^	17.18 ^a^	0.74	0.01
Total PUFA n-3	2.14 ^c^	2.85 ^b^	3.25 ^a^	0.12	0.01
Total PUFA n-6	8.05 ^c^	11.42 ^b^	13.48 ^a^	0.61	0.01
PUFA/SFA ratio	0.19 ^c^	0.28 ^b^	0.35 ^a^	0.03	0.01
MUFA/SFA ratio	0.37 ^c^	0.42 ^b^	0.51 ^a^	0.03	0.01
n-6/n-3 ratio	3.75 ^c^	4.00 ^b^	4.15 ^a^	0.05	0.01

PNF, purple Neem foliage; SFO, sunflower oil; ^a, b, c^ in the same row, there is a statistically significant difference (*p* < 0.05); SEM, standard error of the mean.

## Data Availability

Not applicable.

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
