# Peer review of "Effect of Purple Neem Foliage as a Feed Supplement on Nutrient Apparent Digestibility, Nitrogen Utilization, Rumen Fermentation, Microbial Population, Plasma Antioxidants, Meat Quality and Fatty Acid Profile of Goats"

_animals, 2022, doi:10.3390/ani12212985_

Round 1
Reviewer 1 Report
REVISION
Manuscript ID: animals-1913772
Title: Effect of Purple Neem foliage as a Feed Supplement on Nutrient
Apparent Digestibility, Nitrogen Utilization, Rumen Fermentation, Microbial
Population, Plasma Antioxidants, Meat Quality and Fatty Acid Profile of Goats
Authors: Nittaya Taethaisong *, Siwaporn Paengkoum, Chatsirin Nakharuthai,
Narawich Onjai-uea, Sorasak Thongpea, Boontum Sinpru, Jariya Surakhunthod,
Weerada Meethip, Pramote Paengkou
The manuscript deals with an interesting and current topic and examines the effects on nutritional, metabolic and quality aspects of the product. However, the work has many gaps in the writing phase such as pieces repeated twice, bibliographic citations not consistent with the topic and even some pieces copied from another work published by the same authors. We recommend an assessment of possible plagiarism with appropriate programs for the purpose.
Line 61: The work cited (2) is not consistent with what is reported in the text
Line 74: bibliographic references must be cited in ascending order
Line 79: delete ” –“
Line 117: For an efficient understanding of the methods used and as required by the journal, the authors must report the precise AOAC reference to which the method refers.
Line 135: what does the 60:40 ratio refer to?
Line 142: report the unit of measurement of the components of the diet in the table
Line 146: move the% before =
Line 176: For an efficient understanding of the methods used and as required by the journal, the authors must report the precise AOAC reference to which the method refers.
Line 177: For an efficient understanding of the methods used and as required by the journal, the authors must report the precise AOAC reference to which the method refers.
Line:180-184: these sentences are identical to those reported in the following work: Taethaisong, N., Paengkoum, S., Nakharuthai, C., Onjai-uea, N., Thongpea, S., Sinpru, B., ... & Paengkoum, P. (2022). Consumption of Purple Neem Foliage Rich in Anthocyanins Improves Rumen Fermentation, Growth Performance and Plasma Antioxidant Activity in Growing Goats. Fermentation, 8(8), 373.
Line 186-190: these sentences are identical to those reported in the following work:
Taethaisong, N., Paengkoum, S., Nakharuthai, C., Onjai-uea, N., Thongpea, S., Sinpru, B., ... & Paengkoum, P. (2022). Consumption of Purple Neem Foliage Rich in Anthocyanins Improves Rumen Fermentation, Growth Performance and Plasma Antioxidant Activity in Growing Goats. Fermentation, 8(8), 373.
Line 188: References 23 and 24 are not consistent with the AIA method for measuring digestibility, in fact the method is not described in the works cited.
Line 191-204: these sentences are identical to those reported in the following work:
Taethaisong, N., Paengkoum, S., Nakharuthai, C., Onjai-uea, N., Thongpea, S., Sinpru, B., ... & Paengkoum, P. (2022). Consumption of Purple Neem Foliage Rich in Anthocyanins Improves Rumen Fermentation, Growth Performance and Plasma Antioxidant Activity in Growing Goats. Fermentation, 8(8), 373
Line 290-291: Delete this sentence because it is already written in line 284-285.
Line 306-307: this sentence is repeated, delete
Line 309: The authors must report the statistical model used, for the various parameters, in greater detail and clarity. In particular, for the parameters in which the determinations are performed on the same subjects at different times, the statistical model must take into account the repeated measurements (parameters measured at 0, 2, 4 h) with Anova for repeated measure
Line 333: Authors must report the unit of measurement of the parameters in the table
Line 336-337: the correct transcription of the acronym DNFD is: digestibility of Neutral Detergent Fiber
and the correct transcription of the acronym DADF is: digestibility of Acid Detergent Fiber
Line 345: report the units of measurement of the components in the underlined table
Line 351: Replace blood urea nitrogen (BUN) with rumen urea nitrogen (RUN) as we are talking about rumen fermentation parameters
Line 353: attention to significance: were not different ....... (p> 0.05) not (p<0.01)
Line 355: This statement is not correct if we look at the data in table 4. The only lower value is at 0h but not significant (p <0.37). The only higher and significant pH data is at 4h (p <0.03). Rewrite the result.
In Table 4 in yellow: report the letters indicating differences significant among the averages
In Table 4 in yellow: delete BUN and write RUN
Line 360: Delete and write RUM, rumen urea nitrogen
Line 380: In paragraph 3.6 use the acronyms already clarified in the first lines of the same paragraph.
In Table 12 in yellow: report the unit of measurement
In Table 13 in yellow : report the unit of measurement for all items
Line 411: use acronyms
Line 418-420: in the discussion section, avoid repeating the results already present in the results section.
Line 433: in the discussion section, avoid repeating the results already present in the results section.
Line 462: specify this acronym
Line 473-479: in the discussion section, avoid repeating the results already present in the results section.
Line 499 change BUN in RUM
Line 515: the reference (66) is not related to the statement made in the sentence.
Line 518: the reference (67) is not related to the statement made in the sentence.
Line 525: The bibliographic reference is inconsistent with the discussion delete or find a suitable one
Liu, C.; Sun, J.; Lu, Y.; Bo, Y. Effects of Anthocyanin on Serum Lipids in Dyslipidemia Patients: A Systematic Review and Meta-912 Analysis. PLoS One 2016, 11, e0162089, doi:10.1371/journal.pone.0162089.
Line 585: The bacterium Butyrivibrio fibrisolvens is not a bacterium with proteolytic action but as has long been known and widely verified it is a bacterium that has a hemicellulosolytic, pectinolytic and amylolytic action. (see work cited)
Stern, Marshall D. "Rumen physiology." (2011).
Line 628: eliminate because it refers to the monogastric species
Line 644: In my opinion, melatonin implantation is not comparable to a feed treatment. It is recommended to delete the comparison
Line 700: specify that it is a poultry species not a goat species
Line 707: use the acronym because it is already reported in the MM section
Line 708: use the acronym because it is already reported in the MM section
Line 709:
in work 105 the authors speak of dairy cows not young kids
105. Bryszak, M.; Szumacher-Strabel, M.; El-Sherbiny, M.; Stochmal, A.; Oleszek, W.; Roj, E.; Patra, A.K.; Cieslak, A. Effects of berry 1005 seed residues on ruminal fermentation, methane concentration, milk production, and fatty acid proportions in the rumen and 1006 milk of dairy cows. J Dairy Sci 2019, 102, 1257-1273, doi:10.3168/jds.2018-15322.
Line 711: specify that in the work 106 the authors speak of swine species
Line 716: rewrite the sentence and check consistency with the quoted work (75)

Author Response
Response to Reviewer 1 Comments
Point 1: Line 61: The work cited (2) is not consistent with what is reported in the text
Response 1: The Siamese neem tree is high in antioxidants and has antifungal and antibacterial properties [1].
Point 2: Line 74: bibliographic references must be cited in ascending order
Response 2: Purple Neem foliage has been described as having anthocyanin content. The functional properties of anthocyanins as beneficial coloring compounds have been studied in animals; it modulates the alleviation of heat stress, accelerates antioxidant activity, and stimulates the rumen microbiome [3 ,4 ,5].
Point 3: Line 117: For an efficient understanding of the methods used and as required by the journal, the authors must report the precise AOAC reference to which the method refers.
Response 3: 2.1. Dried Purple Neem Foliage
The fresh Purple Neem foliage of (Azadirachta indica A.) were collected from Su-ranaree University of Technology (SUT) goat and sheep research farm in Nakhon Ratchasima, Thailand, during November, 2020. Authentication of plant material was carried out at the Animal Technology and Innovation, Institute of Agricultural Tech-nology, Suranaree University of Technology laboratory, have been kept. The preserva-tion process of Purple Neem foliage recommended by [15], was conducted until being offered to the animals.
Point 4: Line 135: what does the 60:40 ratio refer to?
Response 4: 60 is roughage source in this study use Pangola (Digitaria eriantha) hay and 40 is concentrate for feeding to the goats from treatment diets.
Point 5: Line 142: report the unit of measurement of the components of the diet in the table
Response 5: Chemical composition (% DM)
Point 6: Line 146: move the% before =
Response 6: Estimated by the equation TDN (%) = DCP + DNFC) + DEE × 2.25 + (DNDF).
Point 7: Line 176: For an efficient understanding of the methods used and as required by the journal, the authors must report the precise AOAC reference to which the method refers.
Response 7: 2.4. Feed and Fecal Sampling
The amount of feed delivered and the number of refused samples were recorded on a daily basis throughout the trial. At the conclusion of the time, feed, refusals, and fecal samples were collected from each individual goat utilizing the full collection procedure. Dry matter (DM), ash, ether extract (EE), and crude protein (CP) were evaluated using [5,16]. Acid detergent fiber (ADF) and neutral detergent fiber (NDF) were determined according to [21,22].
Point 8: Line 177: For an efficient understanding of the methods used and as required by the journal, the authors must report the precise AOAC reference to which the method refers.
Response 8: Acid detergent fiber (ADF) and neutral detergent fiber (NDF) were determined ac-cording to [21,22].
Point 9: Line:180-184: these sentences are identical to those reported in the following work: Taethaisong, N., Paengkoum, S., Nakharuthai, C., Onjai-uea, N., Thongpea, S., Sinpru, B., ... & Paengkoum, P. (2022). Consumption of Purple Neem Foliage Rich in Anthocyanins Improves Rumen Fermentation, Growth Performance and Plasma Antioxidant Activity in Growing Goats. Fermentation, 8(8), 373.
Response 9: 2.5. Urine Sampling Procedures
On the same days as feces, total urine samples were collected in a plastic container treated with 10% sulfuric acid to keep the final pH below 3 and avoid nitrogen (N) loss. At the end of each period, urine samples were collected at about 100 mL of total urine volume, stored, and pooled for total N analysis utilizing [23].
Point 10: Line 186-190: these sentences are identical to those reported in the following work: Taethaisong, N., Paengkoum, S., Nakharuthai, C., Onjai-uea, N., Thongpea, S., Sinpru, B., ... & Paengkoum, P. (2022). Consumption of Purple Neem Foliage Rich in Anthocyanins Improves Rumen Fermentation, Growth Performance and Plasma Antioxidant Activity in Growing Goats. Fermentation, 8(8), 373.
Response 10: 2.6. Apparent Digestibility
The acid insoluble ash (AIA) approach was used to calculate apparent nutrient digestibility (%) as follows: apparent nutritional digestibility (%) = 100 – ((100 × %AIA in diet × %AIA in fecal)/ (%AIA in fecal × %AIA in diet) [24]. All fecal samples were crushed and passed through a 1 mm filter after being oven dried at 65 °C for 72 h be-fore being stored at 4 °C until analysis.
Point 11: Line 191-204: these sentences are identical to those reported in the following work:Taethaisong, N., Paengkoum, S., Nakharuthai, C., Onjai-uea, N., Thongpea, S., Sinpru, B., ... & Paengkoum, P. (2022). Consumption of Purple Neem Foliage Rich in Anthocyanins Improves Rumen Fermentation, Growth Performance and Plasma Antioxidant Activity in Growing Goats. Fermentation, 8(8), 373
Response 11: 2.7. Plasma antioxidant enzyme activity analysis
Blood samples were drawn through jugular venipuncture into a single 10 mL heparin containing vacuum tube at 0, 2, and 4 h following the morning meal on the last feeding week at 08.00. After centrifugation (Allegra R X-30R Centrifuge, Beckman Coulter, USA) at 3500 g for 20 minutes at 4 °C, the blood sample was transferred to a 1.5 mL tube and stored at 20 °C until the antioxidant activity enzymes in the plasma were tested. The levels of total antioxidant (TAC), superoxide dismutase (SOD), gluta-thione peroxidase (GPX), malondialdehyde (MDA), 2, 2-diphenyl-1-picrylhydrazyl (DPPH) scavenging capacity and catalase (CAT) in the plasma were determined using commercial kits (Sigma-Aldrich, USA). The product codes were MAK187, MAK379, MAK437, MAK085, MAK088, and MAK381. All measurement procedures were carried out in accordance with the manufacturer's specifications. As indicated in a technical report, we used an automated enzymatic colorimetric technique on a microplate (96 wells, UV plate), quadruplicate, integrated into a microreader (Varioskan-LUX mul-ti-mode microplate reader, Thermo Scientific, USA). [25].
Point 12: Line 309: The authors must report the statistical model used, for the various parameters, in greater detail and clarity. In particular, for the parameters in which the determinations are performed on the same subjects at different times, the statistical model must take into account the repeated measurements (parameters measured at 0, 2, 4 h) with Anova for repeated measure
Response 12: 2.11. Statistical analyses
The collect data were analyzed using one-way analysis of variance (ANOVA) by SAS version 9.1.3 (SAS Inst. Inc., Cary, NC, USA) model: Yij = µ + τi + εij, where Yij is the observation j (j = 1 to 6) in the treatment, i (i = control, 3%PNF+3%SFO, 6%PNF+3%SFO ), (µ is the overall mean, τi is the effect of the treatment (denoted an unknown param-eter), and εij is the random error with a mean of 0 and variance σ2 in treatment. Dif-ferences between treatment means were determined by Duncan’s new multiple range test [36]. The model included the treatment and sampling time (at 0, 2, 4 h) was used to specify variations using ANOVA. The relative mRNA abundance was calculated using the data's average abundance of the gene as a calibrator, and the data were analyzed using ANOVA. When the p-Value was (p < 0.05), means were split using Tukey's mul-tiple comparisons adjustment.
Point 13: Line 333: Authors must report the unit of measurement of the parameters in the table
Response 13: Apparent Digestibility, % of intake
Point 14: Line 336-337: the correct transcription of the acronym DNFD is: digestibility of Neutral Detergent Fiber and the correct transcription of the acronym DADF is: digestibility of Acid Detergent Fiber
Response 14: DNDF, digestibility of Neutral Detergent Fiber; DADF, digestibility of Acid Detergent Fiber
Point 15: Line 345: report the units of measurement of the components in the underlined table
Response 15: Table 6. Effect of Purple Neem foliage as a Feed Supplement on Nitrogen Balance.
Items |
Control |
3%PNF + 3%SFO |
6%PNF + 3%SFO |
SEM |
p-Value |
N intake, g/d |
11.08c |
12.74b |
13.54a |
0.28 |
0.01 |
N Faces, g/d |
4.75a |
3.56b |
3.25c |
0.18 |
0.01 |
N Urine, g/d |
0.13c |
0.16b |
0.18a |
0.006 |
0.01 |
N digestion, g/d |
4.06c |
6.53b |
8.46a |
0.49 |
0.01 |
N digestion (%) |
37.35c |
48.40b |
62.20a |
2.72 |
0.01 |
N retention, g/d |
4.54c |
6.48b |
8.49a |
0.44 |
0.01 |
N retention (%) |
33.84c |
47.68b |
61.22a |
3.00 |
0.01 |
Point 16: Line 351: Replace blood urea nitrogen (BUN) with rumen urea nitrogen (RUN) as we are talking about rumen fermentation parameters
Response 16: 3.4. Rumen fermentation parameter
The data from the rumen fermentation parameters are shown in Tables 7 and 8, including pH, ammonia nitrogen (NH3-N), rumen urea nitrogen (RUN), acetic acid (C2), propionic acid (C3), butyric acid (C4), acetic acid: propionic acid (C2/C3), and total VFA (TVFA) level. The pH, NH3-N, BUN, C2, C3, C4, and TVFA were not different be-tween the groups at 0 h (p > 0.05). There were significant differences (p < 0.01) in pH, NH3-N, RUN
Point 17: Line 353: attention to significance: were not different ....... (p> 0.05) not (p<0.01)
Response 17: The pH, NH3-N, BUN, C2, C3, C4, and TVFA were not different between the groups at 0 h (p > 0.05).
Point 18: Line 355: This statement is not correct if we look at the data in table 4. The only lower value is at 0h but not significant (p <0.37). The only higher and significant pH data is at 4h (p <0.03). Rewrite the result.
Response 18: Table 7. Effect of Purple Neem foliage as a Feed Supplement on Rumen fermentation.
Items |
Control |
3%PNF + 3%SFO |
6%PNF + 3%SFO |
SEM |
p-Value |
pH |
|||||
0 h |
6.95 |
6.96 |
6.93 |
0.008 |
0.37 |
2 h |
6.75 |
6.76 |
6.77 |
0.008 |
0.58 |
4 h |
6.85 |
6.86 |
6.87 |
0.01 |
0.38 |
Mean |
6.85 |
6.86 |
6.86 |
0.007 |
0.89 |
Ammonia nitrogen mg/dL |
|||||
0 h |
12.43 |
12.65 |
12.52 |
0.07 |
0.47 |
2 h |
12.84 |
13.34 |
14.49 |
0.2 |
0.01 |
4 h |
13.24c |
14.34b |
15.49a |
0.25 |
0.01 |
Mean |
12.84c |
13.44b |
14.17a |
0.15 |
0.01 |
RUN mg/dL |
|||||
0 h |
11.23 |
11.22 |
11.24 |
0.62 |
0.008 |
2 h |
12.35c |
13.42b |
14.35a |
0.22 |
0.01 |
4 h |
13.48c |
14.20b |
15.28a |
0.21 |
0.01 |
Mean |
12.35c |
12.95b |
13.62a |
0.14 |
0.01 |
Point 19: In Table 4 in yellow: report the letters indicating differences significant among the averages
Response 19: a, b, c, in the same row, there is a statistically significant difference averages p < 0.05.
Point 20: In Table 12 in yellow: report the unit of measurement
Response 20: Table 12. Effect of Purple Neem foliage as a Feed Supplement on meat quality characteristics of goats.
Items |
Control |
3%PNF+3%SFO |
6%PNF+3%SFO |
||
SEM |
p-Value |
||||
pH value at |
|||||
1 h |
7.23 |
7.24 |
7.25 |
0.01 |
0.44 |
24 h |
6.85 |
6.84 |
6.83 |
0.006 |
0.79 |
Mean |
7.03 |
7.04 |
7.05 |
0.005 |
0.9 |
Meat Color |
|||||
L* |
31.54 |
31.63 |
31.6 |
0.04 |
0.72 |
a* |
1.75 |
1.76 |
1.77 |
0.01 |
0.76 |
b* |
5.27 |
5.28 |
5.25 |
0.01 |
0.66 |
Drip loss (%) |
5.45a |
4.51b |
3.82c |
0.2 |
0.0001 |
Cooking loss (%) |
32.79a |
29.32b |
24.44c |
1.04 |
0.0001 |
Shear force (N) |
11.70a |
8.93b |
7.28c |
0.52 |
0.01 |
Point 21: In Table 13 in yellow : report the unit of measurement for all items
Response 21: Table 13. Effect of Purple Neem foliage as a Feed Supplement on chemical composition of goats.
Items |
Control |
3%PNF+3%SFO |
6%PNF+3%SFO |
||
SEM |
p-Value |
||||
Moisture, (%) |
76.6 |
76.63 |
76.64 |
0.01 |
0.44 |
Dry matter, (%) |
23.4 |
23.37 |
23.36 |
0.01 |
0.44 |
CP, (%) |
19.42 |
19.43 |
19.45 |
0.009 |
0.6 |
Ash, (%) |
4.16 |
4.17 |
4.18 |
0.007 |
0.48 |
EE, (%) |
10.1 |
10.11 |
10.13 |
0.01 |
0.49 |
Point 22: Line 411: use acronyms
Response 22: 3.8. Fatty acid profile in meat goat
Table 14 shows the result. Saturated fatty acid (SFA), monounsaturated fatty acid (MUFA), and polyunsaturated fatty acid (PUFA) had significant differences (p < 0.01). (PUFA). The saturated fatty acid content was also has reduced with goats fed 6%PNF + 3%SFO group. However, goats fed 6%PNF + 3%SFO were significantly higher (p < 0.01) MUFA and PUFA than that of the other treatments.
Point 23: Line 515: the reference (66) is not related to the statement made in the sentence.
Response 23: Similar to the current work, in [67] the rumen molar concentration of acetate was dis-covered to be affected by dietary fat intake. The development of AA is always accom-panied by the generation of H2 and CO2, but the formation of PA necessitates the for-mation of H2 as PA [66].
Point 24: Line 518: the reference (67) is not related to the statement made in the sentence.
Response 24: Propionic acid levels were increased in goats fed 6%PNF + 3%SFO at 2 and 4 h af-ter feeding, because Purple Neem foliage contains a high anthocyanin content, and anthocyanins may influence glucose metabolism in order to supply more energy to ruminants. Similarly, [66] claims that anthocyanins may be able to influence carbohy-drate metabolism in order to supply more energy to ruminants by raising the fraction of PA. The synthesis of anthocyanin-rich purple corn silage revealed a higher concen-tration of anthocyanins, which seemed to be undigested in the rumen [67].
Point 25: Line 525: The bibliographic reference is inconsistent with the discussion delete or find a suitable one
Response 25: Propionic acid levels were increased in goats fed 6%PNF + 3%SFO at 2 and 4 h af-ter feeding, because Purple Neem foliage contains a high anthocyanin content, and anthocyanins may influence glucose metabolism in order to supply more energy to ruminants. Similarly, [66] claims that anthocyanins may be able to influence carbohy-drate metabolism in order to supply more energy to ruminants by raising the fraction of PA. The synthesis of anthocyanin-rich purple corn silage revealed a higher concen-tration of anthocyanins, which seemed to be undigested in the rumen [67]. Further-more, the kind of VFAs generated in the rumen is determined by the substrate fer-mented, the rumen environment, and the microbial community [68,69]. Enhanced ru-men propionic acid levels are associated with increased insulin secretion, fat deposi-tion, and protein synthesis, while blocking lipolysis and protein breakdown [70]. Ac-cording to the findings of this investigation, the propionic acid concentration in all treatments was within the optimal range of 18-20% [71].
Point 26: Line 585: The bacterium Butyrivibrio fibrisolvens is not a bacterium with proteolytic action but as has long been known and widely verified it is a bacterium that has a hemicellulosolytic, pectinolytic and amylolytic action. (see work cited)
Response 26: Butyrivibrio fibrisolven in this study that was higher in goats fed 6%PNF + 3%SFO at 2 and 4 hours after feeding. Previous study has demonstrated that flavonoid-rich plant extracts improve cell wall degradation as well as the productivity and efficiency of microbial protein production [80]. Anthocyanins are also known to influence the composition and number of ruminal bacterial species by inhibiting or enhancing cer-tain species growth.
Point 27: Line 700: specify that it is a poultry species not a goat species
Response 27: Plant polyphenols can modify fatty acid composition by decreasing oxidation process-es in unsaturated fatty acids [104]. lant polyphenolic flavonoids have been found to protect UFA against oxidants and to activate antioxidant response element (ARE) me-diated gene expression [105].
Point 28: Line 707: use the acronym because it is already reported in the MM section
Response 28: Piper betle powder can enhance conjugated linoleic acid accumulation from biohy-drogenation products, in more PUFA accumulation in the rumen. Furthermore, poly-phenols might modulate rumen PUFA biohydrogenation to increase lipid fraction by lowering rumen skatole production, increasing the beneficial fatty acid content, and raising the product's oxidation stability [40].
Point 29: Line 709:in work 105 the authors speak of dairy cows not young kids
Response 29: Consistent with our findings, [106], proposed that a phenolic-rich plant extract in-creased MUFA and PUFA concentrations in the longissimus dorsi of dairy cows.
Point 30: Line 716: rewrite the sentence and check consistency with the quoted work (75
Response 30: [74], Piper betle powder can enhance conjugated linoleic acid ac-cumulation from biohydrogenation products, in more PUFA accumulation in the ru-men. Furthermore, polyphenols might modulate rumen PUFA biohydrogenation to in-crease lipid fraction by lowering rumen skatole production, increasing the beneficial fatty acid content, and raising the product's oxidation stability [40].

Reviewer 2 Report
The manuscript is interesting scientific contributions to the knowledge of the effect of Purple Neem foliage as a feed supplement on nutrient apparent digestibility, nitrogen utilization, rumen fermentation, microbial population, plasma antioxidants and meat quality in goats. Feeding Purple Neem foliage to growing goats reduces not only environmental problems caused by residue accumulation but also the carbon footprint of animal production; it also improves meat shelf-life stability and quality due to the presence of bioactive compounds, such as anthocyanins, and the effect of phenolic compounds as natural antioxidants in reducing oxidative deterioration of meat during meat production. The paper has high scientific level, the experiment is well designed, the discussion is consistent and the final conclusions are interesting. Therefore, the manuscript may be published in Animal making minor revision:
Suggestions for edition as well as some comments are the following:
Keyword
There are a lot of keywords. Please, select the most important keywords (5-6 keywords)
References
Please update the references. There are a lot of recent references (from 2019 till now) about goat meat quality, which were not included in the manuscript.
Author Response
Point 1: Keyword
There are a lot of keywords. Please, select the most important keywords (5-6 keywords)
Response 1: purple neem foliage; sunflower oil; rumen fermentation; meat quality; fatty acid profile in meat goat
Point 2: References
Please update the references. There are a lot of recent references (from 2019 till now) about goat meat quality, which were not included in the manuscript.
Response 2: 4.7. Meat quality characteristics
This study found that anthocyanin from Purple Neem foliage had no influence on carcass parameters such as slaughter weight, carcass weight, carcass (%), skin, white offal, and red offal regardless of treatment. Because anthocyanin's function might in-crease the rate of meat production. Our results are consistent with those reported by [92].
The pH value is connected to pre-slaughtering, which can also affect meat tex-ture and color [93]. The influence of anthocyanin from Purple Neem foliage on pH at 1 and 24 hours was not observed to be significant, however our pH readings were 6.83-7.25. According to the findings of our study, the pH value is within the normal range and has no effect on meat quality, such as color or shelf life. Previous research has indicated that pH values of cashmere goat meat in the current study (range 7.40 to 6.20) are indicative of good carcass quality [89] and [70] demonstrate that the influence of dietary grape pomace had no effect on pH value among different groups. The pH, on the other hand, is determined by the species of animal, as well as the age of slaughter and storage duration.
Meat color has been recognized as the most essential factor because color is connected with freshness, shoppers have identified meat color for evaluating meat quality. Meat color in this study shows that anthocyanin from Purple Neem leaf has a tendency to reduce meat color (a*) tends to brown the meat, implying that the oxymyoglobin to metmyoglobin conversion stage and the lipid peroxidation interaction were engaged in meat discoloration [94]. Our findings are consistent with those of [70,95], who found that dietary grape pomace had no effect on meat color coordinates (L*, a*, and b*) because anthocyanin, a dietary antioxidant, can be better incorporated into cellular membranes containing oxidation susceptible phospholipids than the superficial contact made by antioxidants added postmortem. The color of meat is determined by a variety of individual elements and their interactions; however, chevon has been observed to be lighter in color and more redness than lamb, owing to the decreased intramuscular fat content of goat carcasses [96].
In the current experiment, the effect of anthocyanin from Purple Neem foliage on drip loss and cooking loss was considerable across all treatments. However, drip loss and cooking loss were decreased in goats given 6%PNF + 3%SFO compared to other treatments. Cooking loss, according to [97], is more reliant on the final pH and the cooking condition. Similar to [96,98], our investigation indicated that goat muscles had lower cooking loss than lamb muscles. According to the current study, goats fed 6%PNF + 3% SFO had decreased drip loss and cooking loss. Juiciness of meat is strong-ly connected to intramuscular lipids and moisture content of the meat, and lean meat contains around 75% water [99]. It concludes that anthocyanin is a natural antioxidant that may be transferred to the muscle where, in conjunction with the native defense system, it can counteract the action of pro-oxidants and protect meat water loss, qual-ity, and nutritious value of goat meat.
Shear force many factors impact muscle value, including diet, stress before to and after slaughter, animal age and breed, muscle type, and cooking procedure. It has been observed that changes in collagen fiber organization in connective tissue, as well as a decrease in soluble collagen concentration, are important variables deter-mining shear force value throughout animal development [100]. Tenderness of goat meat is frequently acceptable to consumers [101]. Shear force was shown to be significantly varied across all treatments in this investigation. Previous research has demonstrated that anthocyanin-rich black cane silage (AS) reduces shear force [3]. However, according to the current study, goat fed 6%PNF + 3%SFO has lower shear force com-pared to other treatments because anthocyanin is a natural phenolic compound with health-improving characteristics, such as their potential as a viable alternative to an-tibiotics and synthetic growth boosters in the creation of sustainable animal feed, can serve as an antioxidant and has been demonstrated to be effective in avoiding milk and meat oxidation [102]. Nonetheless, dietary anthocyanin bioactive compounds obtained from Purple Neem foliage may result in a decrease in shear force by enhancing juiciness and meat softness by reducing cooking loss and drip loss. Similar to the findings of [98], treatment with wine grape pomace reduced shear force in the lamb's longissimus dorsi muscle. The chemical composition of the goat meat in this investigation was within the range of previously published values [76], as predicted, was unaffected by dietary treatments. Our findings are consistent with those of [103], who re-ported no effect on the protein, fat, or moisture content of suckling lamb flesh [104], When thyme and rosemary were introduced to sheep diets, no changes in muscle lamb chemical composition.
4.8. Fatty acid profile in meat
Plant polyphenols can modify fatty acid composition by decreasing oxidation processes in unsaturated fatty acids [105]. plant polyphenolic flavonoids have been found to protect UFA against oxidants and to activate antioxidant response element (ARE) mediated gene expression [106]. In the current study, goats fed a diet of 6%PNF + 3%SFO lowered the percentage of saturated fatty acids in goat meat because anthocyanin inhibited the process of biohydrogenation in the rumen. [19], discovered that include flavonoid-rich plants in the diet of dairy goats might reduce milk individual SFA and total SFA concentrations while increasing the UFA profile. In contrast, the content of MUFA and PUFA was greater in goats fed a diet containing 6%PNF + 3%SFO. Consistent with our findings, [107], proposed that a phenolic-rich plant ex-tract increased MUFA and PUFA concentrations in the longissimus dorsi of dairy cows. The anthocyanin addition in ruminant feed benefits in the prevention of USF [108]. Characterizing the individual and combined impacts of particular phytochemicals in Purple Neem foliage with anthocyanin concentration will have biohydrogenation products that might assist increase the quantities of human-health-promoting PUFAs in ruminant meat. [74,95 Piper betle powder can enhance conjugated linoleic acid ac-cumulation from biohydrogenation products, in more PUFA accumulation in the ru-men. The addition of fermented Saccharina japonica and Dendropanax morbifera to meat has been shown to enhance PUFA levels [109]. Furthermore, polyphenols might modulate rumen PUFA biohydrogenation to increase lipid fraction by lowering rumen skatole production, increasing the beneficial fatty acid content, and raising the product's oxidation stability [40].

Round 2
Reviewer 1 Report
Table 4. Nutrient intake g DM/d replace with “Nutrient intake”
Line 689: In order to give precise information to readers, the authors are suggested to add to the sentence "Plant polyphenols can modify fatty acid composition by decreasing oxidation processes in unsaturated fatty acids in broilers [105]. “
Line 690: to add “Plant” polyphenolic ....
Author Response
Point 1: Table 4. Nutrient intake g DM/d replace with “Nutrient intake”
Response 1: Table 4. Effect of Purple Neem foliage as a Feed Supplement on Feed Intake.
Items |
Control |
3%PNF+3%SFO |
6%PNF+3%SFO |
||
SEM |
p-Value |
||||
Feed intake |
|||||
gDM/d |
855.60c |
971.56b |
1055.11a |
24.68 |
0.01 |
%BW |
3.07c |
3.15b |
3.27a |
0.02 |
0.01 |
g/kgBW0.75 |
43.96c |
59.04b |
61.64a |
2.09 |
0.01 |
Nutrient intake |
|||||
OMI, g/d |
1500.20c |
1517.76b |
1542.41a |
4.63 |
0.01 |
CPI, g/d |
41.40c |
46.63b |
50.56a |
1.01 |
0.01 |
EEI, g/d |
8.57c |
10.59b |
11.56a |
0.34 |
0.01 |
NDFI, g/d |
1269.14a |
1154.54b |
1096.63c |
19.16 |
0.01 |
ADFI, g/d |
771.83a |
725.56b |
673.52c |
10.75 |
0.01 |
Point 2: Line 689: In order to give precise information to readers, the authors are suggested to add to the sentence "Plant polyphenols can modify fatty acid composition by decreasing oxidation processes in unsaturated fatty acids in broilers [105]. “
Response 2: Plant polyphenols can modify fatty acid composition by decreasing oxidation processes in unsaturated fatty acids in broilers [105].
Point 3: Line 690: to add “Plant” polyphenolic ....
Response Plant polyphenolic flavonoids
